# Computational 3D histological phenotyping of whole zebrafish by X-ray histotomography

Yifu Ding[1,2,3], Daniel J Vanselow[1,2], Maksim A Yakovlev[1,2], Spencer R Katz[1,2,3], Alex Y Lin[1,2], Darin P Clark[4], Phillip Vargas[5], Xuying Xin[1,2], Jean E Copper[1,2], Victor A Canfield[1,2], Khai C Ang[1,2], Yuxin Wang[6], Xianghui Xiao[7], Francesco De Carlo[8], Damian B van Rossum[1,2], Patrick La Riviere[5], Keith C Cheng[1,2]*

[1]The Jake Gittlen Laboratories for Cancer Research, Penn State College of Medicine, Hershey, United States; [2]Division of Experimental Pathology, Department of Pathology, Penn State College of Medicine, Hershey, United States; [3]Medical Scientist Training Program, Penn State College of Medicine, Hershey, United States; [4]Center for In Vivo Microscopy, Duke University, Durham, United States; [5]Department of Radiology, The University of Chicago, Chicago, United States; [6]Imaging Group, Omnivision Technologies, Inc., Santa Clara, United States; [7]National Synchrotron Light Source II, Brookhaven National Laboratory, Upton, United States; [8]Advanced Photon Source, Argonne National Laboratory, Lemont, United States

*For correspondence:
kcheng76@gmail.com

**Abstract** Organismal phenotypes frequently involve multiple organ systems. Histology is a powerful way to detect cellular and tissue phenotypes, but is largely descriptive and subjective. To determine how synchrotron-based X-ray micro-tomography (micro-CT) can yield 3-dimensional whole-organism images suitable for quantitative histological phenotyping, we scanned whole zebrafish, a small vertebrate model with diverse tissues, at ~1 micron voxel resolutions. Micro-CT optimized for cellular characterization (histotomography) allows brain nuclei to be computationally segmented and assigned to brain regions, and cell shapes and volumes to be computed for motor neurons and red blood cells. Striking individual phenotypic variation was apparent from color maps of computed densities of brain nuclei. Unlike histology, the histotomography also allows the study of 3-dimensional structures of millimeter scale that cross multiple tissue planes. We expect the computational and visual insights into 3D cell and tissue architecture provided by histotomography to be useful for reference atlases, hypothesis generation, comprehensive organismal screens, and diagnostics.
DOI: https://doi.org/10.7554/eLife.44898.001

## Introduction

Histology has been used for over a century to visualize cellular composition and tissue architecture in millimeter- to centimeter-scale tissues from diverse multicellular organisms (*Virchow, 1860*). Its diagnostic power is dependent on the detection and description of changes in cell and tissue architecture. Normal and abnormal cytological features indicative of physical, inflammatory, and neoplastic causes of disease are readily distinguished using established staining procedures (*Kumar et al., 2015*).

**eLife digest** Diagnosing diseases, such as cancer, requires scientists and doctors to understand how cells respond to different medical conditions. A common way of studying these microscopic cell changes is by an approach called histology: thin slices of centimeter-sized samples of tissues are taken from patients, stained to distinguish cellular components, and examined for abnormal features. This powerful technique has revolutionized biology and medicine. But despite its frequent use, histology comes with limitations. To allow individual cells to be distinguished, tissues are cut into slices less than 1/20th of a millimeter thick. Histology's dependence upon such thin slices makes it impossible to see the entirety of cells and structures that are thicker than the slice, or to accurately measure three-dimensional features such as shape or volume.

Larger internal structures within the human body are routinely visualized using a technique known as computerized tomography, CT for short – whereby dozens of x-ray images are compiled together to generate a three-dimensional image. This technique has also been applied to image smaller structures. However, the resolution (the ability to distinguish between objects) and tissue contrast of these images has been insufficient for histology-based diagnosis across all cell types. Now, Ding et al. have developed a new method, by optimizing multiple components of CT scanning, that begins to provide the higher resolution and contrast needed to make diagnoses that require histological detail.

To test their modified CT system, Ding et al. created three-dimensional images of whole zebrafish, measuring three millimeters to about a centimeter in length. Adjusting imaging parameters and views of these images made it possible to study features of larger-scale structures, such as the gills and the gut, that are normally inaccessible to histology. As a result of this unprecedented combination of high resolution and scale, computer analysis of these images allowed Ding et al. to measure cellular features such as size and shape, and to determine which cells belong to different brain regions, all from single reconstructions. Surprisingly, visualization of how tightly the brain cells are packed revealed striking differences between the brains of sibling zebrafish that were born the same day.

This new method could be used to study changes across hundreds of cell types in any millimeter to centimetre-sized organism or tissue sample. In the future, the accurate measurements of microscopic features made possible by this new tool may help us to make drugs safer, improve tissue diagnostics, and care for our environment.

DOI: https://doi.org/10.7554/eLife.44898.002

Despite its power, histology has practical limitations in throughput and quantitative phenotyping of cell and tissue volume and shape. Physical sectioning introduces tissue loss and distortions that compromise complete reconstruction of tissues from serial histology sections (*Arganda-Carreras et al., 2010*). The technical demands and physical properties of paraffin blocks limit slice thickness to ~5 µm, leading to incomplete sampling and imperfect visualization of elongated structures such as vessels and large cells that extend beyond the section thickness. As a result, only a small fraction of any given tissue sample is studied in histology. Moreover, it is intractable to generate complete sets of sections for large numbers of whole organisms (as needed for a genetic or chemical screen) and impractical to align them for volumetric analysis. A routine method for histological phenotyping that avoids these problems and enables comprehensive three-dimensional (3D) analysis of whole organisms would transform large-scale studies.

X-ray micro-tomography (micro-CT) is a potential means of achieving complete, 3D histological phenotyping for large numbers of specimens. Micro-CT is commonly used to study hard, mineralized tissues like bones and fossils (*Donoghue et al., 2006*; *Ketcham and Carlson, 2001*). Soft-tissue imaging typically requires contrast-enhancing, heavy-metal stains such as osmium tetroxide, iodine, phosphotungstic acid (PTA), or gallocyanin-chromalum (*Betz et al., 2007*; *Metscher, 2009a*; *Metscher, 2009b*). Phase-based synchrotron imaging of unstained samples allows the study of development in live whole *Xenopus* embryos over time (*Moosmann et al., 2013*) and of dense human cerebellar tissue (*Töpperwien et al., 2018*). Commercial micro-CT devices have been used to interrogate soft-tissue structure in diverse stained specimens (*Badea et al., 2008*; *Batiste et al.,*

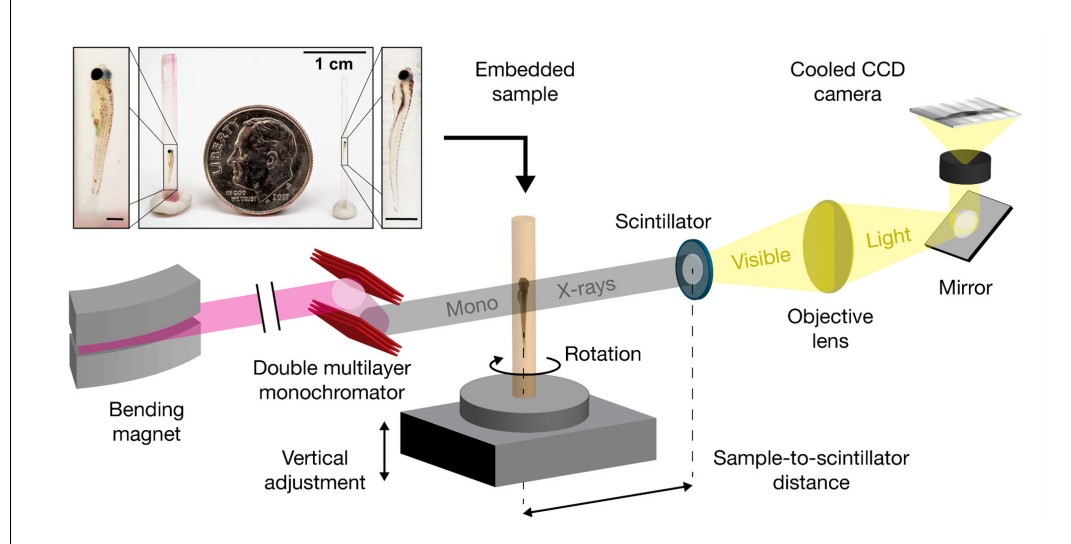

**Figure 1.** Schematic for synchrotron X-ray micro-tomography of whole zebrafish. Quasi-parallel X-rays from beamline 2-BM-B are used to acquire projection images of an intact, fixed, and PTA-stained whole zebrafish. Total imaging time is ~20 min per monochromatic acquisition (sample-to-scintillator distance = 30 mm) and ~20 s per pink-beam acquisition (sample-to-scintillator distance = 25 mm). Each fish requires 3 to 5 acquisitions. The top inset shows the relative sizes of a juvenile (left), a larva (right), and a dime (diameter = 17.9 mm). Fish are shown embedded in acrylic resin with the polyimide tubing removed. Removal of the polyimide tubing is necessary for natural color light photography, but not for successful X-ray image acquisition. Scale bars in the specimen insets are 1 mm.

DOI: https://doi.org/10.7554/eLife.44898.003

The following figure supplements are available for figure 1:

**Figure supplement 1.** X-ray energy optimization for synchrotron micro-CT.
DOI: https://doi.org/10.7554/eLife.44898.004

**Figure supplement 2.** Sample-to-scintillator distance selection for synchrotron X-ray micro-tomography.
DOI: https://doi.org/10.7554/eLife.44898.005

**Figure supplement 3.** Comparison of image quality between monochromatic and polychromatic X-rays for synchrotron micro-CT.
DOI: https://doi.org/10.7554/eLife.44898.006

---

*2004*; *Cheng et al., 2016a*; *Metscher, 2009a*; *Staedler et al., 2013*), but these devices utilize polychromatic, low-flux X-ray tube sources that limit resolution and throughput. Synchrotron X-ray sources have monochromatic, high-flux beams (*Winick, 1994*) that allow rapid imaging of mm-scale samples such as insects (*Betz et al., 2007*; *Mizutani et al., 2013*; *van de Kamp et al., 2018*), vertebrate embryos (*Khonsari et al., 2014*; *Raj et al., 2014*), zebrafish (*Seo et al., 2015*), and mouse somatosensory cortex (*Dyer et al., 2017*). Micro-CT of larger samples such as whole mice or whole mouse organs have lacked adequate tissue contrast and/or resolution (~10 $\mu m^3$) for histological phenotyping (*Busse et al., 2018*; *Hsu et al., 2016*). Conversely, nano-CT enables imaging at higher resolution (~100 $nm^3$) but does not provide sufficient field-of-view for mm-scale samples. To our knowledge, no existing method has the combination of throughput, resolution, field-of-view, and soft-tissue contrast necessary for whole-organism 3D phenotypic screens that are inclusive of histopathological evaluation.

Here, we present a 3D quantitative histological analysis of features of whole fixed and stained zebrafish, at sub-micron resolution, using synchrotron micro-CT optimized for soft tissue differentiation (histotomography). We chose zebrafish, the largest established vertebrate model to fit in our field-of-view (juveniles are <3 mm in diameter), to determine conditions for attaining a degree of tissue contrast required for histopathological analysis across a range of tissues. X-ray histotomography as developed here is pan-cellular, like histology, allowing for the creation of histology-like virtual sections, but unlike histology, can be used to study millimeter-scale phenotypes involving convoluted

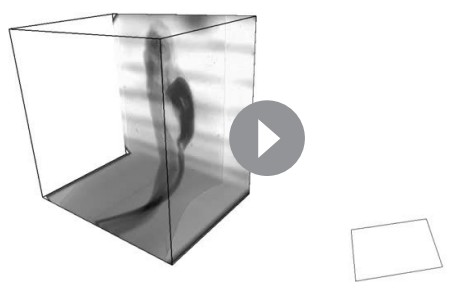

**Video 1.** X-ray Projection and Sinogram. This video demonstrates the relationship between multi-angle projection data and sinograms, which are used in reconstruction to backproject single axial slices that are combined to generate a volume (online viewing available from, https://youtu.be/MjobFBLc5m8). Best if viewed at highest quality setting.
DOI: https://doi.org/10.7554/eLife.44898.007

and branched structures such as gills, gut, vessels, and nerve tracts. Histotomographic images provide a mechanism for quantitative histopathological analysis that facilitates the objective and reproducible study of volume, shape, and texture of cells across organisms and tissue types. Histotomography's potential throughput and analytical power may be applied to computational histological phenotyping of whole organisms to probe the diversity of organismal tissue architecture and relationships between genotype, environment, and microanatomy (*Austin et al., 2004*; *Cheng et al., 2016b*; *Varshney et al., 2013*).

## Results

### Whole-Organism synchrotron X-ray histotomography

Synchrotron based micro-CT was chosen based on two demands of phenome projects: the potential to achieve resolutions and contrast required for histological evaluations, and high-throughput potential for phenotyping whole organisms. Fine-tuning of X-ray energies and bandwidth, and adjustment of sample-to-scintillator distance, were used to optimize volumetric reconstructions of optically-opaque zebrafish at isotropic resolutions (equal in all three dimensions) for histopathological interpretations. Synchrotron X-ray flux is orders of magnitude more brilliant than commercial tube sources, allowing imaging times short enough for high sample-number imaging projects (*Winick, 1994*).

Micro-CT studies were performed on the sector two bending magnet, hutch B (2-BM-B) of the Advanced Photon Source at Argonne National Laboratory for both single-energy and multi-energy acquisitions (*Figure 1*). Beam energies for monochromatic imaging were optimized for contrast-to-noise ratio across a range of sample diameters and concentrations of tungsten (*Figure 1—figure supplement 1*). A beam energy of 13.8 keV was used for the smaller, larval samples and 16.2 keV for juveniles due to their greater thickness (see Materials and methods). Sample-to-scintillator distance was adjusted to optimize the magnitude of phase contrast-based edge enhancement (*Figure 1—figure supplement 2*). A 30 mm sample-to-scintillator distance provided sufficient edge enhancement to achieve histology-like contrast. For whole-organism imaging, the zebrafish were vertically translated to capture segments over the full length of the sample. Each zebrafish segment was reconstructed and the multiple segments stitched into a single 3D volume (*Video 1*).

Large-scale phenotyping studies, such as those focusing on series of mutants or chemical exposures, require short imaging times to facilitate throughput. Monochromatic acquisitions took ~20 min,~36 fold faster than acquisition by commercial tabletop sources (e.g.,~720 min for Xradia 500 series machines) (*Metscher, 2009b*). The use of polychromatic 'pink-beam' increases X-ray flux, which greatly reduces exposure times. Each pink-beam acquisition took ~20 s,~60 fold faster than monochromatic acquisition and ~2,000 fold faster than commercial sources. Monochromatic acquisitions have better image quality than pink-beam, as defined by signal-to-noise ratio and pixel intensity profile (*Figure 1—figure supplement 3*). Notably, no image degradation over years of repetitive synchrotron imaging was detected (*Lin et al., 2018*). This high degree of sample stability across multiple scans suggests the potential use of pink-beam pre-screening followed by higher resolution monochromatic reacquisition.

### Digital 3D zebrafish at cell resolution

Digital zebrafish were reconstructed from projections after staining all tissues with PTA. The similarity of digital slices to conventional histological outputs is demonstrated by representative transverse (axial), sagittal, and coronal cross-sections for larval (4 days post-fertilization, dpf) and juvenile (33 dpf) zebrafish imaged at 0.743 and 1.43 $\mu m^3$ isotropic voxel resolution, respectively (*Figure 2*).

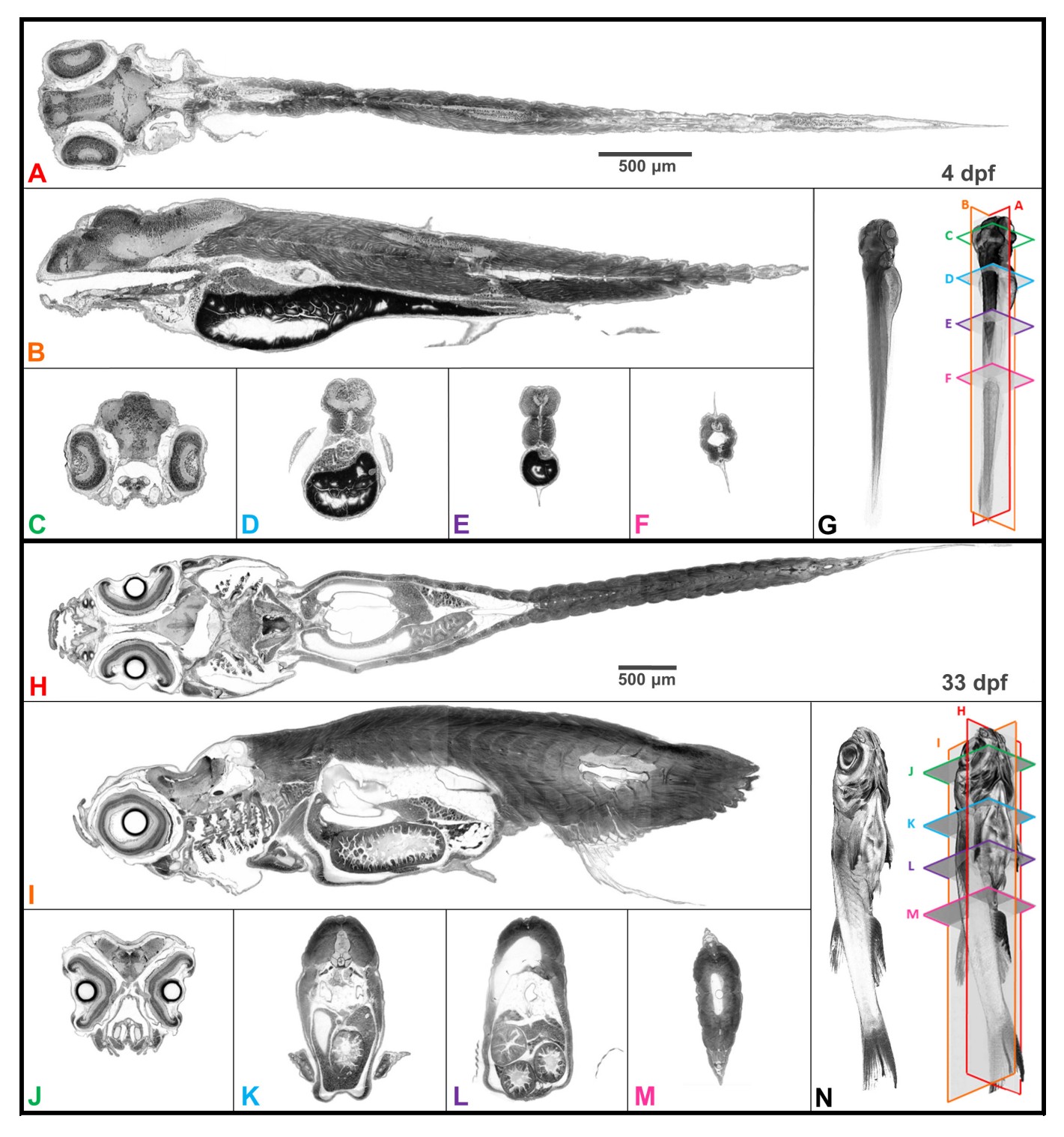

**Figure 2.** Whole organism imaging of PTA-stained zebrafish at cell resolution enables histology-like cross sections. Coronal (A, H), sagittal (B, I), and axial (C–F, J–M) cross sections of 4 dpf larval (A–G) and 33-dpf juvenile (H–N) wild-type zebrafish acquired using synchrotron X-ray micro-tomography at 0.743 µm$^3$ and 1.43 µm$^3$ isotropic voxel resolution, respectively. 3D volume renderings (G, N) show the cross-sections in relation to the whole organism. In contrast to histology, the cross sections are a single voxel in thickness and can be obtained in any plane (including oblique cuts) after imaging. Complete cross-sections in the orthogonal directions for both fish are provided (*Videos 2* and *3*). Images are presented to match histological convention of dark cell nuclei (higher attenuation is darker).
DOI: https://doi.org/10.7554/eLife.44898.008

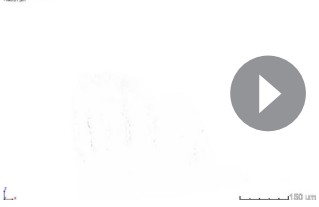

**Video 2.** Larval Flipbook. This video shows full cross-sections of the larval (five dpf) zebrafish from the sagittal orientation (0.743 micron slice thickness) for histology-like phenotyping and qualitative analysis (online viewing available from, https://youtu.be/hyyZu2_75Qc). Intensity histogram of the dataset was inverted and locally adjusted to better discern originally faint or overlapping structures. Best if viewed at highest quality setting.
DOI: https://doi.org/10.7554/eLife.44898.009

Volume renderings illustrate the orientation of individual planes of section in 3D. Full sets of cross-sections of the zebrafish in the sagittal orientation illustrate one way to use micro-CT data to phenotype full tissue volumes (*Videos 2–3*).

Notably, the z-axis resolution of histotomography, presented here at 0.74 or 1.43 microns, is superior to the ~5 micron slice thickness of histology's conventional paraffin sections. Increased z-axis resolution decreases the likelihood of two separate nuclei being scored as one when they are crowded, as is the case in the retinal and brain neurons of larval zebrafish. The problem of overlapping nuclei in the z-axis makes it difficult to accurately count nuclei from traditional histological images. Our x-y histotomographic resolution is comparable with that of optical microscopy using a 10X objective lens (*Figure 3*). When direct comparisons between histotomography and histology are desired, maximum intensity projections (MIPs) totaling ~5 microns in thickness can be created from image stacks. Just as in hematoxylin and eosin (H and E) stained sections from ~3.5 mm-long larval (five dpf) (*Figure 3A*) and ~1 cm-long juvenile (33 dpf) zebrafish (*Figure 3C*), cell and tissue types can be recognized histologically from five micron MIPS of age-matched fish (*Figure 3B and D*). As an internal measure of resolution, we calculated the spacing of striations in skeletal muscle fibers encircling the larval swim bladder (*Figure 3—figure supplement 1*). The average of 293 measures = 2.16 μm (SD = 0.55 μm) is comparable with published values (*Burghardt et al., 2016*; *Dou et al., 2008*).

## Forms of analysis of zebrafish microanatomy enabled by histotomography

While both traditional histology and histotomography have the resolution needed to distinguish cellular features in 2D slices, only the latter is able to reveal elongated, complex 3D tissue structures such as vessels, nerve tracts, and bones. These advantages are particularly well-illustrated by the study of gill architecture (*Figure 3C and D*, insets). Histotomography makes it possible to interrogate primary and secondary lamellae from multiple angles without spatial distortion, revealing the delicate leaf-like structure of gill filaments on any pharyngeal arch and even bulges in epithelial cells caused by their nuclei (*Video 4*). Whole animal histotomographic reconstructions thus provide a level of organismal and anatomical context that is not practical using histology alone.

The isotropic nature of histotomography enables reslicing in any plane of section without distortion. Dynamic reslicing (cutting the same volume digitally in multiple planes) allows the study of either longitudinal, perpendicular, or other planar virtual sections of the gut at cellular resolution (*Figure 4*). As far as we are aware, the ability to generate full sets of cross-sections of convoluted structures (such as the gut of juvenile zebrafish) for histological phenotyping within a single modality is unprecedented. This approach can be applied to any convoluted structure requiring thorough evaluation.

Virtual sectioning and 3D rendering can be used to study any tissue in the fish, as illustrated for the neuronal cells in the eye, cartilaginous elements of the notochord, the squamous patch of the dorsum of the pharynx, nucleated red blood cells in the heart, and the pneumatic duct and

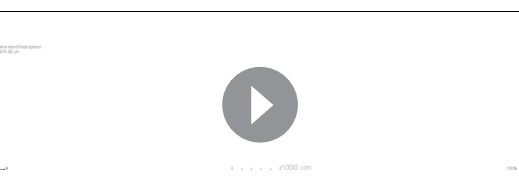

**Video 3.** Juvenile Flipbook. This video shows full cross-sections of the juvenile (33 dpf) zebrafish from the sagittal orientation (1.4 micron slice thickness) for histology-like phenotyping and qualitative analysis (online viewing available from, https://youtu.be/gyxmFJGU1RY). Intensity histogram of the dataset was inverted and locally adjusted to better discern originally faint or overlapping structures. Best viewed at highest quality setting.
DOI: https://doi.org/10.7554/eLife.44898.010

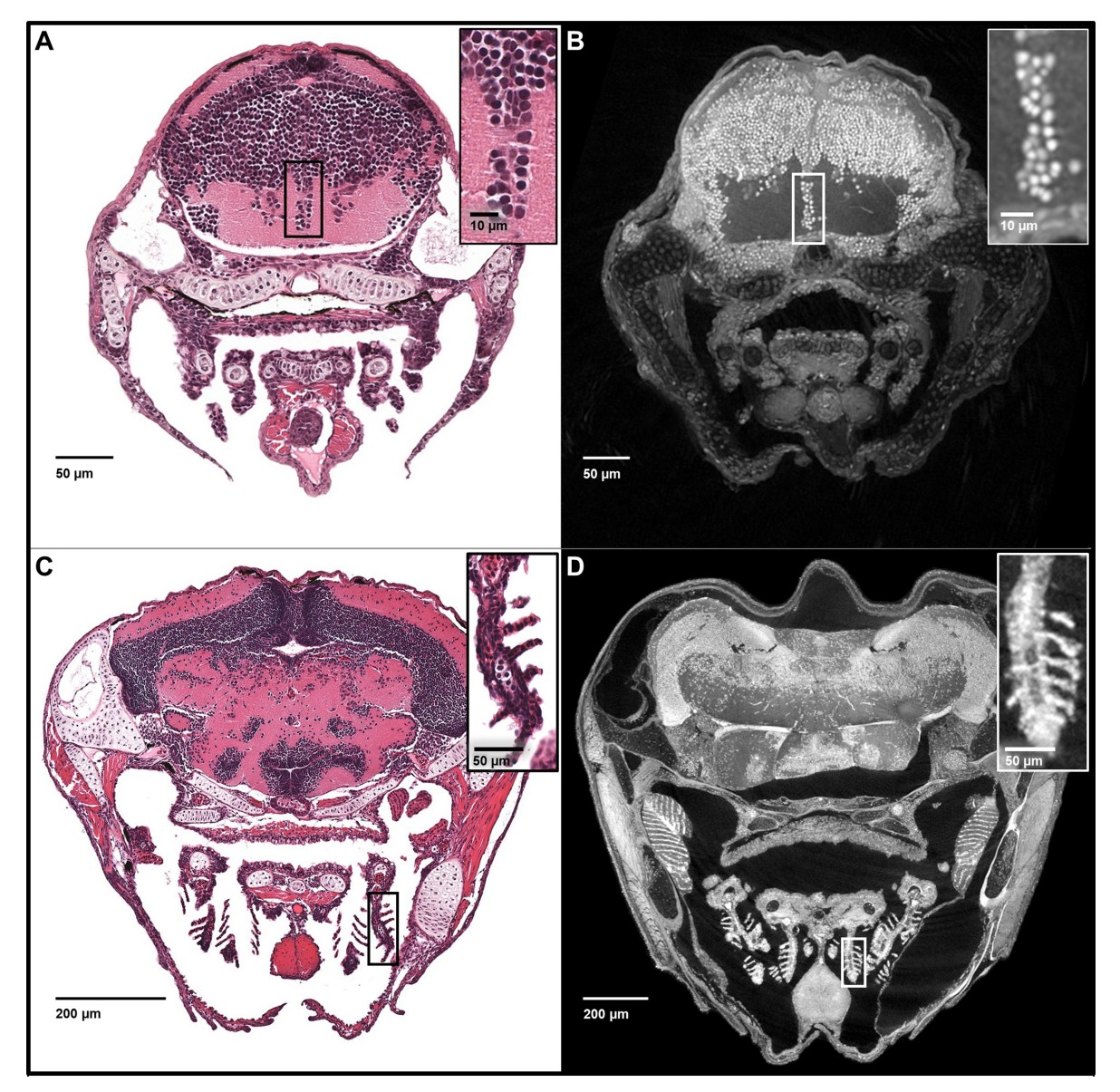

**Figure 3.** Five micron thick maximum intensity projections of synchrotron X-ray histotomographic images resemble histology. *Top:* Comparison of larval (five dpf) zebrafish obtained from a 5 μm thick histology section and micro-tomography data (A and B, respectively). The larval micro-tomograph is 0.743 μm³ in isotropic voxel resolution, and a maximum intensity projection (MIP) of 7 slices, totalling 5.20 μm, resembling the appearance of ~5 μm thick histological sections. *Bottom:* Comparison of juvenile (33 dpf) zebrafish i5 μm histology section and micro-tomographic MIPs (C and D, respectively). Juvenile scan data is of 1.43 μm³ isotropic voxel resolution, and a MIP of 3 slices, totalling 4.29 μm thickness, is shown. Insets show detail of brain cell nuclei (**A–B**) and delicate gill structure (**C–D**). The images demonstrate the near histological resolution of X-ray histotomography. While natural variation in the size of specific features is observed in age matched fish (panel C length = 7.8 mm, panel D length = 10 mm) individual histological features are consistent.

DOI: https://doi.org/10.7554/eLife.44898.011

The following figure supplement is available for figure 3:

**Figure supplement 1.** Bands of skeletal muscle wrapping the air bladder of larval zebrafish allow characterization of resolution.
DOI: https://doi.org/10.7554/eLife.44898.012

goblet cells of the intestine of whole juvenile (33 dpf) and larval (five dpf) specimens in *Figure 5* and *Figure 6*, respectively. In sum, our digital zebrafish allow tissues and cell nuclei to be visualized across organ systems, including the integumentary, hematopoietic, respiratory, genitourinary,

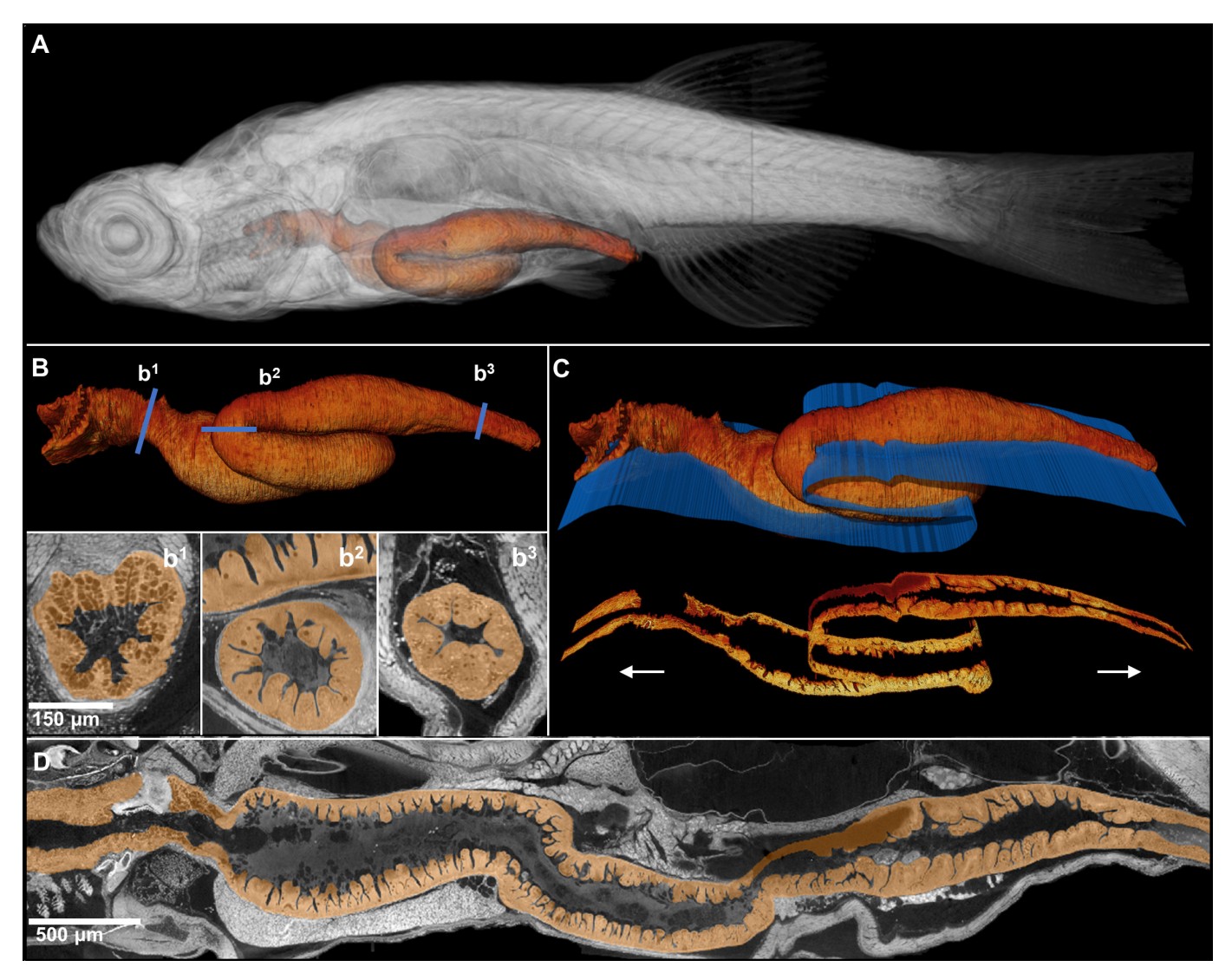

**Figure 4.** Comprehensive histological cross-sectioning of convoluted structures in juvenile zebrafish. A 3D rendering of a whole juvenile (33 dpf) zebrafish is presented with highlighted gastrointestinal (GI) tract (**A**). The GI tract exemplifies a convoluted structure for that can be unraveled for histological analysis. The isolated GI tract is displayed (**B**). Isotropic resolution of data permits virtual slicing at any angle (**B1–B3**) without a decrease in resolution, allowing comprehensive histology-like visualization despite its tortuous nature. A spline-based reslicing method for visualizing serpentine organs across their entire path is also shown (**C**). A nonlinear cutting plane (blue) follows the structure of interest, allowing the cut to render a structure's total length onto a single plane (**D**).

DOI: https://doi.org/10.7554/eLife.44898.014

musculoskeletal, gastrointestinal, cardiovascular, nervous and sensory systems (*Figure 5—figure supplement 1*, *Video 5*).

## Detection of histopathological features in whole-fish histotomography

Successful histopathological analysis requires the detection of subtle differences in the characteristics of specific micron-scale structures. To probe the potential ability of X-ray histotomography to detect subtle histopathological change in a whole organism, we used a mutant, *huli hutu* (*hht*), that is known to show a range of subtle to obvious histological changes across all cell types and tissues (*Mohideen et al., 2003*) (*Figure 6*, *Video 6*). Reconstructions of larval *hht* at 0.743 μm³ voxel resolution allowed us to detect all of *hht*'s known histological changes, including nuclear fragmentation

(karyorrhexis) that is associated with cell death, nuclear atypia (increased size, deviation from typical ovoid shape, and irregular nuclear membrane contour) in the gut, and tissue degeneration. Nuclear fragments are commonly two microns or less in diameter. We were able to reliably establish the absence of an easily missed, micron-scale structure, the pneumatic duct, since every virtual slice of entire fish is available at sub-micron resolution. This structure cannot be reliably assessed by histology on account of its small size and tortuous shape. Age-matched wild-type and *hht* fish (2, 3, 4 and 5 dpf) can be examined, using our web-based data sharing interface, to visually track progression of the histopathological changes characteristic of *hht* (*Table 1*). Additionally, computational detection of nuclei allows global assessment of the density of nuclei across the entire animal (second half of *Video 6*).

## Distribution of cell nuclei reveals phenotypic variation between the brains of larval zebrafish

Due to the combination of histotomography's field-of-view and resolution, we are able to compute features of individual cells and pattern of cells throughout the whole organism. Anatomic pathologists use cytological characteristics in qualitative tissue assessments (*Al-Abbadi, 2011*; *Cheng and Bostwick, 2002*). Quantitative assessment of these morphological attributes is best-accomplished in 3D, bringing added precision in distinguishing disease states from the range of normal tissue and cellular architecture. To evaluate the power of computational phenotyping enabled by histotomography, we combined regional segmentation with automated cell detection for characterization of the zebrafish brain (*Figure 7*).

For example, brain cell nuclei can be distinguished from other stained objects based on shape (elongation) and volume. Elongation is measured as the ratio of the major axis over the minor axis of a 3D object. As anticipated, computational measurements of shape and volume of manually segmented red blood cells (RBCs) and motor neurons (*n* = 20 each) were different, in agreement with morphological differences that are readily apparent in histology (*Figure 8A*). In addition, RBCs and motor neurons are more elongated than typical brain nuclei, which are more spherical. Specifically, RBCs are typically flat and oval, while motor neurons are more 'tear drop' shaped. Other cell differences include the volume, with motor neurons being larger than typical brain nuclei. The PTA staining patterns of the cytoplasm of RBCs and motor neurons are different in appearance from those of other cells. From the measurements shown, it is apparent that our current resolution is sufficient to readily distinguish cell types.

We have assigned brain nuclei to major brain regions including the olfactory epithelia, telencephalon, diencephalon, hypothalamus, mesencephalon, metencephalon, myelencephalon, white matter, and spinal cord by cross-atlas registration (*Raj et al., 2018*; *Ronneberger et al., 2012*; *Wullimann and Mueller, 2005*). In addition, cell nuclei were automatically detected from head scans of intact fish using a supervised learning approach with a random forest classifier. Three 75 μm³ regions of neural cell nuclei were manually annotated and compared against classifier results for location and counts. The $F_1$ score, optimized for probability threshold to balance precision and recall, showed ~90% correspondence between manual and automated detections (*Figure 7—figure supplement 1*). The distribution of cell nuclei in our samples corresponded well with those seen in 54 nm thick transmission electron microscopy sections (*Hildebrand et al., 2017*) (*Figure 7—figure supplement 2*).

Whole-animal histotomography enables quantitative analyses of both individual cells and groups of cells. The average number of nuclei in the brain region out of 5 larvae studied was 75,413 (SD = 8,547) (*Table 2*). This number corresponds well to the ~80,000 reported brain cell ROIs in similarly-aged zebrafish imaged with light

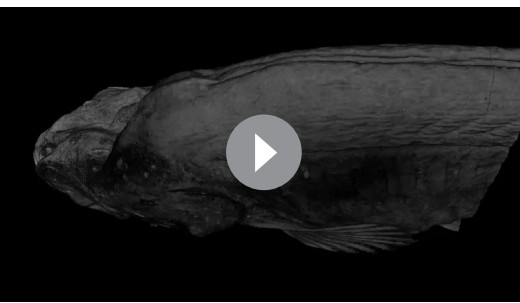

**Video 4.** Gill Flyover. The gill structure is visualized in a whole juvenile (33 dpf) zebrafish in order to demonstrate the ability of soft tissue synchrotron micro-CT to resolve the complex structure of 3D tissues in detail (online viewing available from, https://youtu.be/16sZpZZj9GU). Best viewed at highest quality setting.

DOI: https://doi.org/10.7554/eLife.44898.013

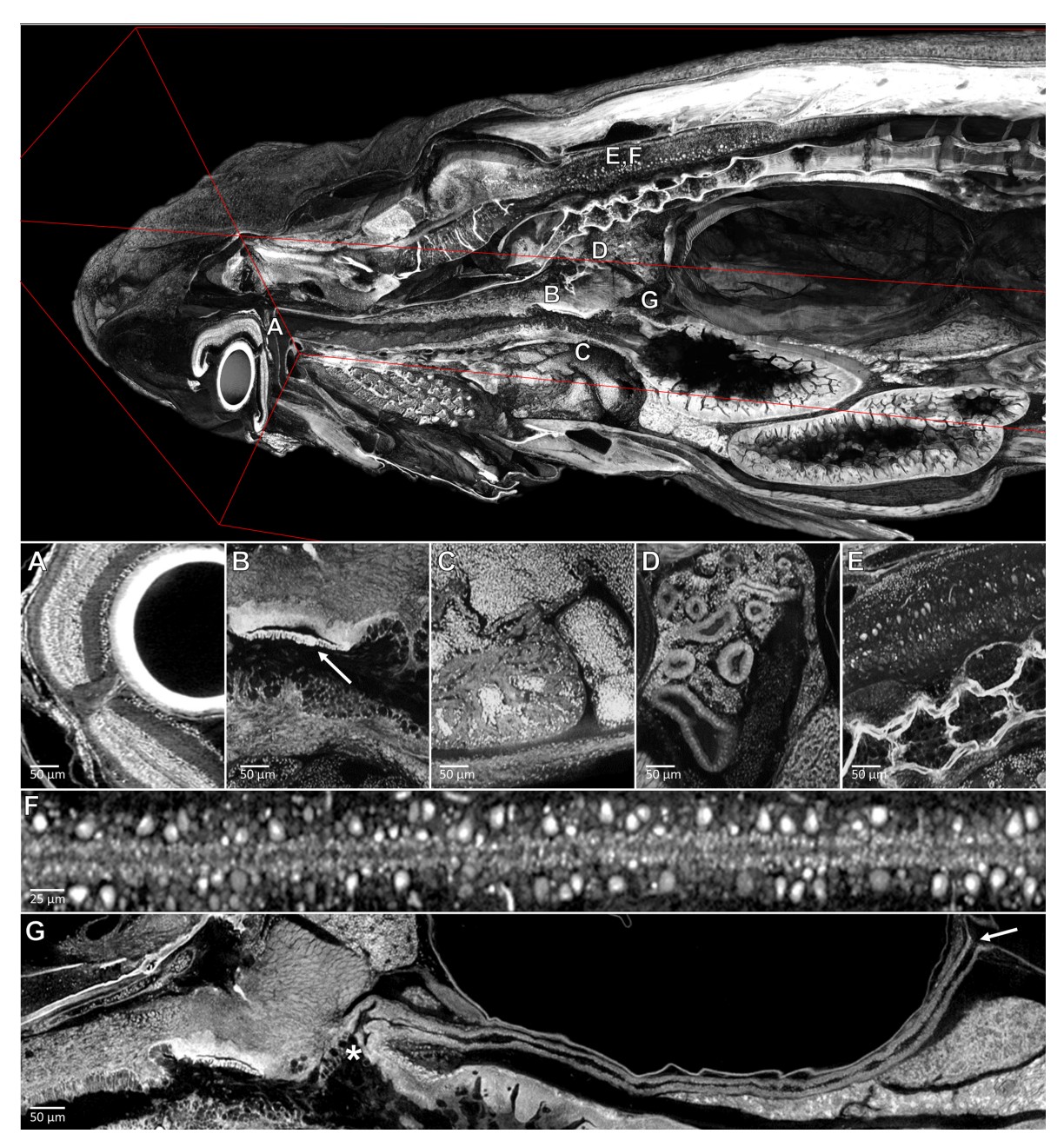

**Figure 5.** Pan-cellular staining and variable-thickness views allow for characterization of 3D microanatomical features. *Top:* Cutout visualization of a juvenile (33 dpf) zebrafish stained with PTA showing detail in many soft tissue structures. *Bottom*: Cell types and structures that can be visualized include neuronal cells in the eye (**A**), cartilaginous rudiments of the squamous patch dorsum (arrow) of the pharynx (**B**), nucleated red blood cells and heart chambers (**C**), nephrons of the kidney (**D**), brain nuclei and motor neurons in spinal nerve cord (**E, F**). Curved multiplanar slicing, as used for *Figure 4D*, was used to display the full length of the pneumatic duct (proximal and distal ends noted by an '*' and arrow, respectively) (**G**). Panels A and E represent individual slices (1.43 μm in thickness) while panels B, C, D represent maximum intensity projections of 5 μm thick sections to visualize larger 3D structures. F represents a 7 μm thick projection.
DOI: https://doi.org/10.7554/eLife.44898.015

The following figure supplement is available for figure 5:

**Figure supplement 1.** ViewTool, a web-based, digital, multi-planar histology interface.
DOI: https://doi.org/10.7554/eLife.44898.016

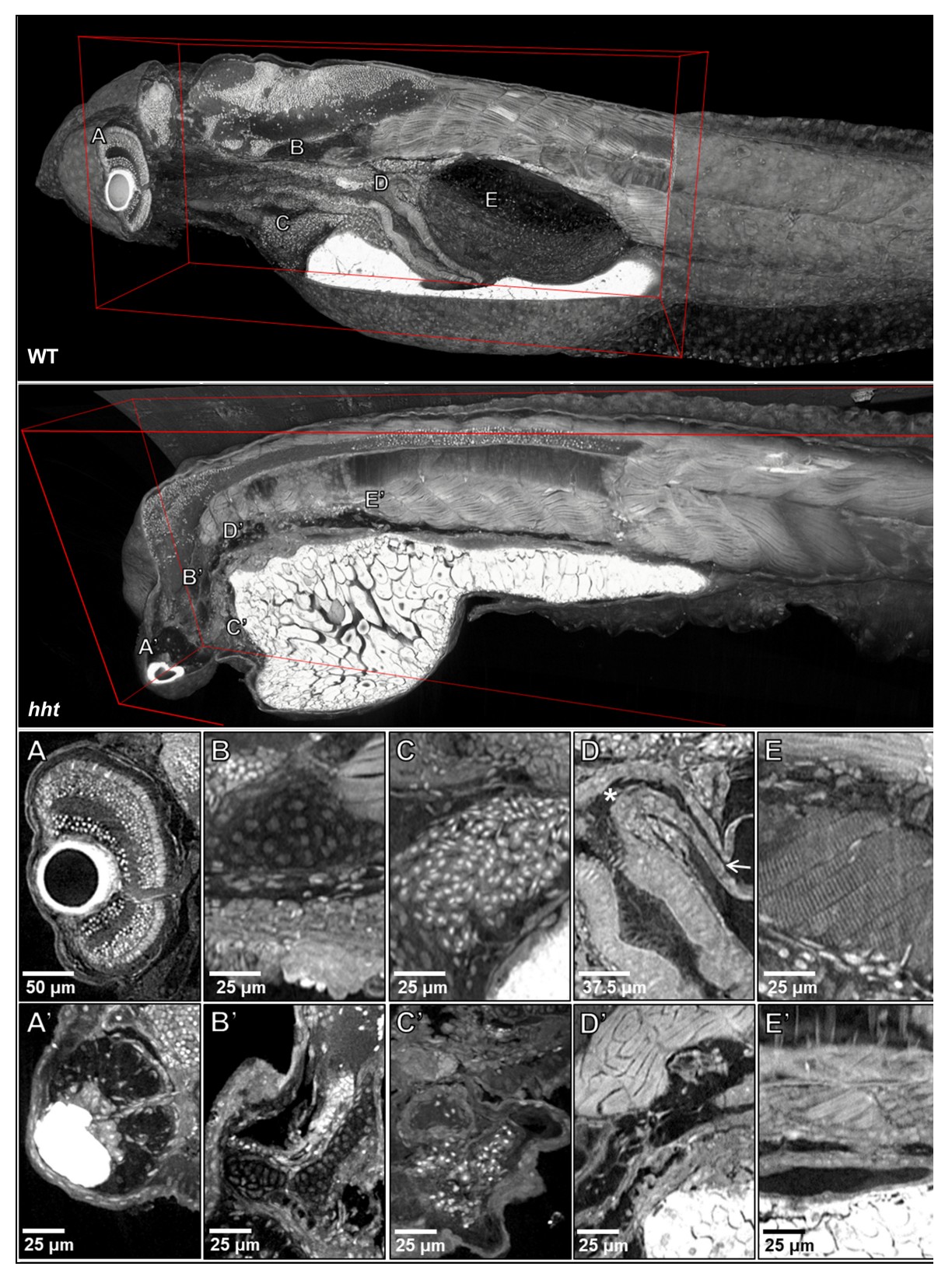

**Figure 6.** Pan-cellular staining and variable-thickness views allow for characterization of 3D pathological features in wild-type larval and *huli hutu* mutant specimens. *Top:* Cutout visualization of both wild-type and *huli hutu* larval (five dpf) zebrafish stained with PTA showing detail in many soft tissue
*Figure 6 continued on next page*

*Figure 6 continued*

structures. *Bottom*: Cell types and structures that can be visualized include neuronal cells in the eye (**A**), cartilaginous rudiments of the squamous patch on the dorsal (arrow) pharynx (**B**), nucleated red blood cells (**C**), intact pneumatic duct (* to arrow) and goblet cells in the gut (**D**), and cross-striations of bands of muscles encircling the swim bladder (**E**). Panels A and D represent individual slices (0.743 µm in thickness) while B, C, E represent maximum intensity projections of 5 µm thick sections to visualize larger 3D structures. Compared to age matched wild-type larval zebrafish (*top*), the number of neuronal cells in the eye are markedly reduced (**A'**), chondrocytes appear cytologically normal, but formation of cartilaginous structures is markedly reduced (**B'**), the myocardium is thickened and, as is evident from a survey through all the sections of the heart (a single slice shown here) contains a markedly reduced number of nucleated red blood cells, consistent with anemia and abnormal hematopoiesis (**C'**). We are able determine the absence of the pneumatic duct and swim bladder in *hht* due to the ability to scan through the full volume of the sample. D' shows degenerate tissue and E' other tissues where those organs normally lie. A', B' and D' represent individual slices (0.743 µm in thickness) while C' and E' represent maximum intensity projections of 5 µm thick sections to visualize structures of larger dimension.

DOI: https://doi.org/10.7554/eLife.44898.017

sheet microscopy (*Chen et al., 2018*). In contrast to the general agreement of size and elongation distributions of brain cell nuclei between individual fish (*Figure 8*), nuclear density varied between brain regions, as reflected by cell density distributions and 3D patterning (*Figure 9*). Notably, heat-maps of cell densities revealed obvious differences between five wild-type siblings. Comparatively, fish 1 and 2 exhibit a shift towards lower cell density as compared with fish 3, 4, and 5, which correlates with the larger brain volumes of fish 1 and 2 (*Tables 2–3*). These differences in cell density may be explained by differences in exact developmental stage.

## Discussion

Large-scale studies of the effects of genes and environment on phenotype (phenome projects) are ideally comprehensive and quantitative in nature, covering all cell and tissue types across length scales. We pursued synchrotron micro-CT to enable the vision of computationally phenotyping small organisms in 3D, at a throughput and resolution that is compatible with phenome projects (*Cheng et al., 2011*). Other groups have utilized X-ray micro-CT for quantitative morphometric analyses of juvenile and adult teleost fish (*Babaei et al., 2016*; *Seo et al., 2015*; *Weinhardt et al., 2018*). Histological analysis across cell types would add significant powerful to phenotypic screens, but would require higher resolution and contrast-to-noise ratios than available at the time. This need motivated us to systematically optimize details of sample preparation, sample mounting, imaging acquisition settings, X-ray optics, image processing, and visualization parameters to make highly informative histological signatures apparent.

Histotomography offers resolving power over more than four orders of magnitude, providing both anatomical and cellular detail from single images that encompass, for zebrafish, the whole organism. High-throughput adaptations would be necessary for whole-organism chemical and genetic phenome projects. Genetic phenomics may involve, for example, the study of mutants for each zebrafish protein-encoding gene (>26,000) and non-coding functional element. By our estimates, screening 10,000 zebrafish mutants in replicate would require ~20 years with a monochromatic synchrotron source and, if fully optimized, <1 year using pink-beam X-rays. High-throughput primary screens by pink-beam could be followed by higher-contrast monochromatic imaging of samples that show evidence of phenotypic change. The fact that some 70% of human protein-coding genes have at least one zebrafish orthologue makes this throughput potential relevant to disease modeling and the probing of human gene function (*Howe et al., 2013*). The unprecedented depth of zebrafish tissue

**Video 5.** Juvenile Flythrough. Tissue structure of a juvenile (33 dpf) zebrafish stained with PTA and imaged with synchrotron micro-CT are visualized through a core-like cut from the center of the digitized sample (online viewing available from, https://youtu.be/FNG-NwXsGHA). Best viewed at highest quality setting.
DOI: https://doi.org/10.7554/eLife.44898.018

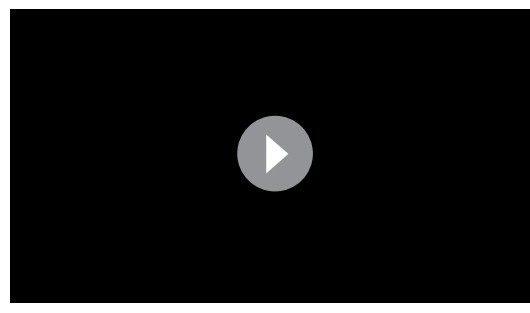

**Video 6.** *Huli hutu* Larval Flipbook and Cell Detection. This video shows full cross-sections of the *huli hutu* mutant larval (five dpf) zebrafish from the sagittal orientation (0.743 micron slice thickness) for histology-like phenotyping and qualitative analysis (online viewing available from, https://youtu.be/KbVnamIMPiA). Neuronal cell nuclei probabilities of the mutant are also shown along with a wild-type comparison. The intensity histogram of the dataset was inverted and locally adjusted to enhance the visibility of faint or overlapping structures. Best viewed at highest quality setting.

DOI: https://doi.org/10.7554/eLife.44898.020

phenotyping enabled by histotomography has the potential to enrich conclusions from mouse phenome projects (*Dickinson et al., 2016*; *Hsu et al., 2016*). Plastic embedding of specimens, while not essential for synchrotron micro-CT, adds sample stability (*Lin et al., 2018*) f image re-acquisition with improved micro-CT implementations.

Optical imaging modalities such as light sheet fluorescence microscopy are ideal for in vivo studies (*McDole et al., 2018*), and resolve 3D sub-micron features. Fluorescence-based imaging, however, depends on sample transparency, and for optically more opaque samples, dissection and/or physical slicing to maintain resolution without diffraction or loss of signal (*Chung et al., 2013*; *Hama et al., 2011*; *Susaki et al., 2014*; *Watson et al., 2017*). Image quality is compromised in the presence of significant melanin pigmentation, even in transparent samples with diameters of more than about a millimeter. The molecular specificity of fluorescence-based phenotyping is poorly suited for large-scale screens that require phenotyping across all cell types.

Three-dimensional tissue imaging methods based on ultrathin serial tissue sections or block-face imaging have ultrastructural cellular resolution, but becomes intractable for large scale studies of specimens larger than ~1 cubic millimeter in minimum dimension. Serial block-face scanning electron microscopy has been used to image and reconstruct a mouse neocortex (*Kasthuri et al., 2015*) and the larval zebrafish brain at nanometer-scale ($56.4 \times 56.4 \times 60$ nm$^3$ resolution from 16,000 sections) (*Hildebrand et al., 2017*). These studies made it possible to define precise neurological and circulatory relationships across the entire brain. However, 3D reconstructions based on serial sectioning are time-consuming enough to be infeasible

**Table 1.** ViewTool database.

Scans listed are available for viewing on ViewTool at http://3D.fish and are PTA stained unless otherwise noted. Raw scans are available on request. Note: The wild-type larval five dpf samples, used for quantitative analysis, were imaged on the same day. *Only the cranial section (head) is presented.

| Zebrafish specimen | Age (dpf) | Segments | 12-bit projections (GB) | 32-bit reconstructions (GB) |
|---|---|---|---|---|
| Larval (wildtype) | 2 | 2 | 25.2 | 64 |
| | 3 | 2 | 25.2 | 64 |
| | 4 | 3 | 37.8 | 96 |
| | 5 | 3 | 37.8 | 96 |
| | 5 | 1* | 12.6 | 32 |
| | 5 | 1* | 12.6 | 32 |
| | 5 | 1* | 12.6 | 32 |
| | 5 | 1* | 12.6 | 32 |
| Larval (*huli hutu*) | 2 | 1* | 12.6 | 32 |
| | 3 | 1* | 12.6 | 32 |
| | 4 | 1* | 12.6 | 32 |
| | 5 | 2 | 25.2 | 64 |
| Juvenile | 33 | 5 | 63.0 | 160 |

DOI: https://doi.org/10.7554/eLife.44898.019

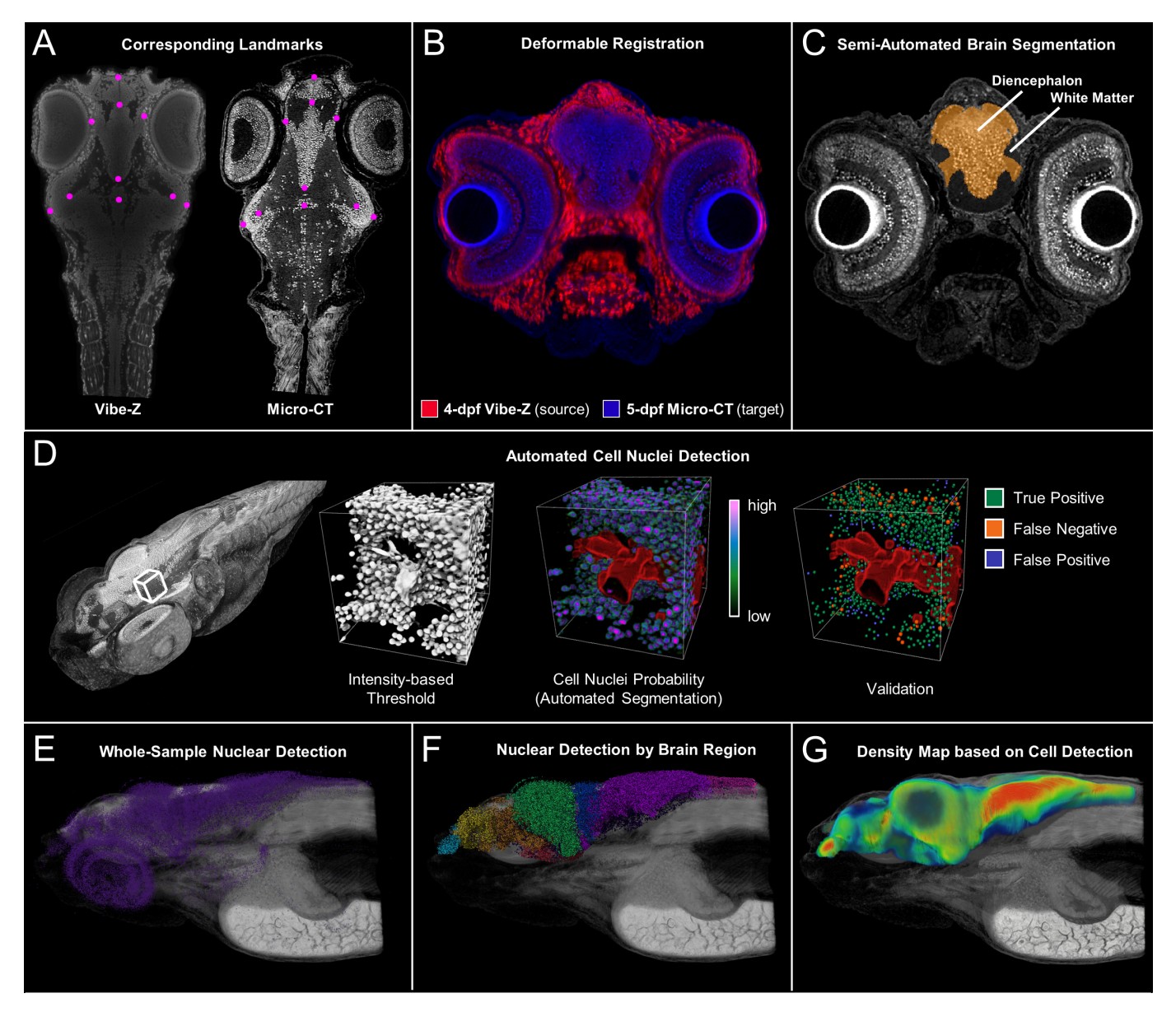

**Figure 7.** Cross-atlas registration and neuronal cell nuclei detection for assignment of automatically detected cells to major brain regions in developing zebrafish. Developing larval (five dpf) zebrafish brains were divided into 10 quantifiable anatomically relevant regions by registering sample brains to the Vibe-Z volume-based zebrafish brain atlas and referencing to an immunohistochemistry BrdU based neuroanatomy atlas. Sample landmark selection is shown for equivalent slices between the ViBE-Z atlas data and a five dpf micro-CT for registration-based segmentation (A). Overlay of ViBE-Z atlas embryo (red) over five dpf data (blue) post-registration (B). Semi-automated brain segmentation presented for a five dpf sample in its anatomic context in a representative slice (C). Neuronal cell nuclei detection and validation using a supervised random forest classifier (D). Nuclear detection visualized as a point map for a typical five dpf sample in its anatomic context (E). Semi-automated brain segmentation overlaid onto classifier-based nuclei detection (F). Brain density was calculated by counting every nucleus within a ~ 22 micron (30 voxel) radius surrounding each voxel of the brain (G)..This dimension waschosen because it covers about 5 cell diameters, an estimate of the width of a small brain region.
DOI: https://doi.org/10.7554/eLife.44898.021

The following figure supplements are available for figure 7:

**Figure supplement 1.** Validation of automated detection of neuronal cell nuclei in larval (five dpf) zebrafish.
DOI: https://doi.org/10.7554/eLife.44898.022

**Figure supplement 2.** Comparison of object detection between micro-CT (five dpf) and transmission electron microscopic (5.5 dpf) sections.
DOI: https://doi.org/10.7554/eLife.44898.023

**Table 2.** Cell nuclei counts in different five dpf zebrafish brain regions.

Major divisions of the zebrafish brain and the number of cell nuclei in each division (counts) are shown across five fish. Mean total cell nuclei count is 75,413 and relative standard deviation ±11.3%.

| Major brain divisions | Brain region cell nuclei (Counts) | | | | |
|---|---|---|---|---|---|
| Specimen number | 1 | 2 | 3 | 4 | 5 |
| Olfactory Epithelium | 855 | 398 | 1103 | 1158 | 1257 |
| Telencephalon | 3696 | 4735 | 4576 | 3704 | 4570 |
| Diencephalon | 6323 | 6076 | 7390 | 7718 | 8134 |
| Hypothalamus | 2096 | 2342 | 3392 | 3047 | 2680 |
| Mesencephalon | 17,377 | 15,993 | 21,161 | 21,448 | 22,275 |
| Metencephalon | 876 | 1025 | 1012 | 2865 | 1713 |
| Myelencephalon | 27,899 | 28,691 | 30,991 | 37,769 | 34,766 |
| White Matter | 5633 | 5955 | 5810 | 4524 | 5181 |
| Spinal Cord | 1408 | 1570 | 1850 | 1712 | 2312 |
| Total | 66,163 | 66,785 | 77,285 | 83,945 | 82,888 |

DOI: https://doi.org/10.7554/eLife.44898.026

for interrogating all tissues in the numbers of specimens required for toxicology or genetic screens. For example, a multi-beam scanning electron microscope, imaging a single cubic mm of tissue at 20 $nm^3$ resolution can require ~3 months of continuous imaging (*Dyer et al., 2017*).

Ideal comprehensive anatomical phenotyping would include quantitative characterization of tissue architecture and cellular features in addition to measures of larger features such as organ size, vascular structure, nerve structure, and body shape. Striking individual phenotypic variation was revealed here by the computational measurement of cell density across larval zebrafish brains. This finding illustrates how histotomography's unique combination of field-of-view and resolution can be useful for distinguishing individual from experimentally-determined phenotypic variation.

Imaging and analysis of multiple specimens at multiple ages is necessary to establish a knowledge of normal statistical variation in gross and microscopic anatomy across organ systems that is needed for automated detection of 'abnormal' phenotypes. Comprehensive computational phenotyping will thus require the development of detailed 3D atlases. Whole specimen histotomography datasets are well-suited for automated organ and cellular level detection that can be used as a scaffold for morphological lifespan atlases, capturing the extent of normal phenotypic variation at cellular, tissue, and organismal scales. The pan-cellular nature of histotomography makes it ideal for cross-referencing with images from focused (e.g. fluorescence-based) modalities involving transgenic or whole-mount stained organisms; this would involve fixation, staining and histotomographic imaging of the fluorescently studied specimens. Conversely, the addition of tissue-, cell-, and/or protein-specific micro-CT stains would enable targeted analyses and cross-atlas comparisons of explicit constituents within the full context of a scanned sample or organism (*Metscher and Müller, 2011*). Further optimization will be needed to resolve a key characteristic of proliferating cells: mitotic chromosomes. Condensed chromosomes are about one micron in diameter, requiring true 0.5 micron pixel/voxel resolution for detection. Increasing field-of-view beyond ~2.7 mm (used for histotomography of juveniles at 1.4 micron isotropic voxel resolution) will be needed to image of the full width of mature zebrafish, and will be facilitated by larger imaging chip arrays.

The work shown here approaches the ideal of comprehensive and quantitative phenotyping for tissue samples and model organisms that lie in the mm-to-cm length scale, such as the zebrafish. Indeed, X-ray histotomography allows for full volume imaging of replicate samples at cellular resolution, distinguishing neighboring cells and resolving nuclear morphologies in optically opaque larval and juvenile zebrafish specimens. Extending this work to *Drosophila*, *C. elegans*, and *Arabidopsis*, and to tissue samples from larger organisms such as mouse and human would enable cross-species phenotypic analyses, leading to a more universal understanding of tissue architecture and morphological phenotype.

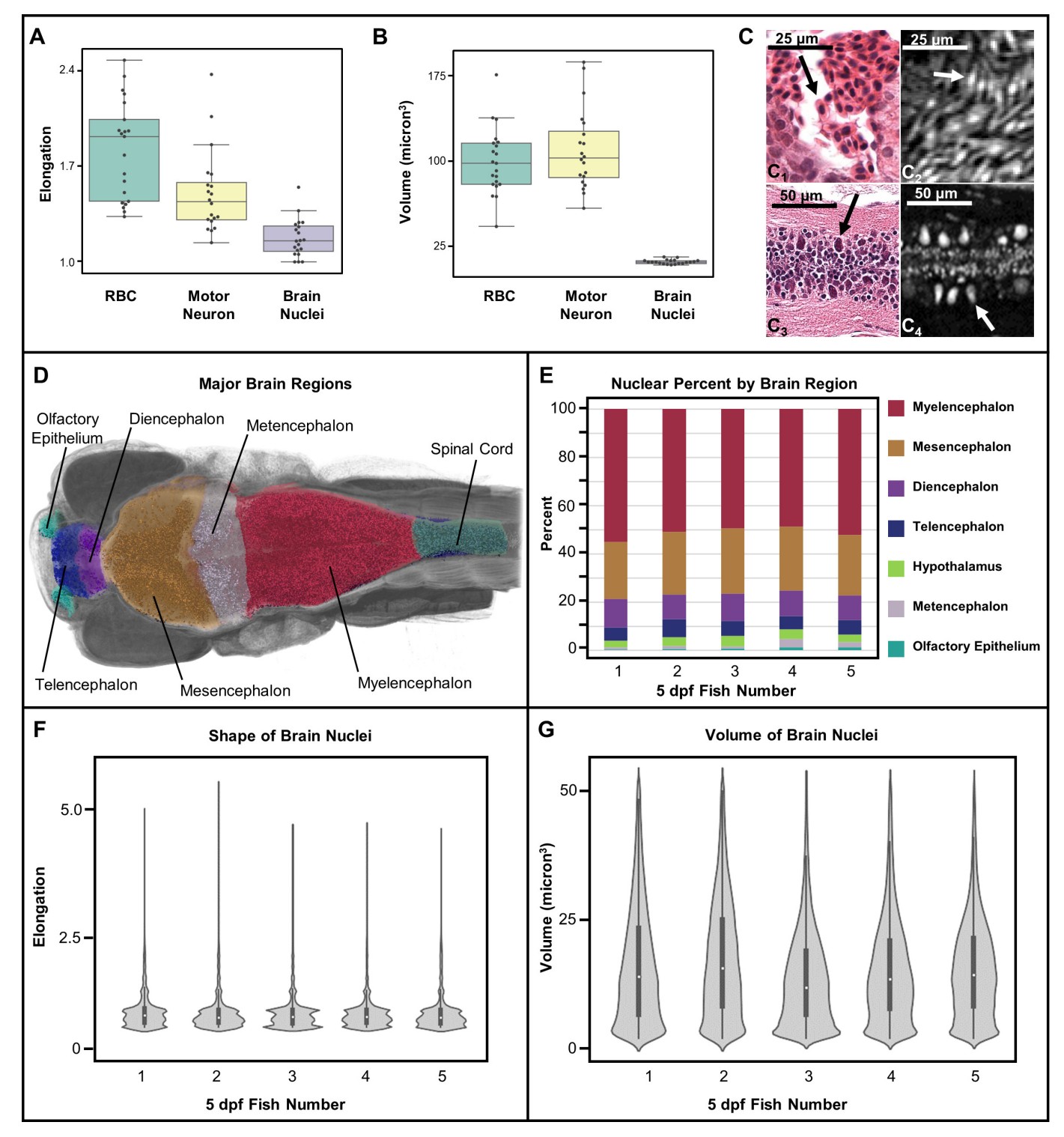

**Figure 8.** Measurement of shape and volume of brain nuclei. Shape and volume of manually detected brain cell nuclei varies between red blood cells (RBCs) and motor neurons (n = 20 cells each) (**A, B**). The mean elongation was 1.9, 1.3, and 1.1 for RBCs, motor neurons, and brain nuclei, respectively. The mean volume was 99, 102, and 13 $\mu m^3$ for RBCs, motor neurons, and brain nuclei, respectively. These releative values are consistent with what is visually apparent in both histology (**C1 and C3**) and histotomography (**C2 and C4**). Volumes of motor neuron and erythrocytes include both nuclei and cytoplasm. Differences in the distribution of brain nuclei were also observed. Cell nuclei were also identified across whole zebrafish samples via a manually trained classifier and segregated by registering brain regions for five five dpf samples (**D**). The regions identified through anatomical

*Figure 8 continued on next page*

*Figure 8 continued*

landmarks include olfactory epithelium, telencephalon, diencephalon, mesencephalon, metencephalon, myelencephalon, hypothalamus, spinal cord, and white matter. The proportion of cells per brain region as percentage of total cell counts, agree in rank order between samples (**E**). Computed elongation and volumes showed no significant difference between individual fish (**F, G**).

DOI: https://doi.org/10.7554/eLife.44898.024

To make our data accessible to users with standard computational resources, we have developed an open-access, web-based data sharing platform, ViewTool, that allows 2D and 3D visualizations of full sample volumes. Histotomographic data are also ideal for real-time visualization using virtual reality technologies such as syGlass (*Pidhorskyi et al., 2018*). Beyond data exploration, the rate of development of methods and tools for computational analysis can be enhanced by long-term accessibility of histotomographic data through an internationally available portal analogous to those used for DNA or transcriptome-based resources. Repositories of high-resolution images from a full complement of human and other organismal tissues would facilitate cross-correlations between model system and human phenotypes (*Regev et al., 2017*; *Rozenblatt-Rosen et al., 2017*). Such a resource would have the potential to increase analytical precision, sensitivity, reproducibility, and data sharing as we address important questions across basic and clinical sciences.

# Materials and methods

**Key resources table**

| Reagent type (species) or resource | Designation | Source or reference | Identifiers | Additional information |
|---|---|---|---|---|
| Strain (*Danio rerio*) | Wild-type Ekkwill | ZFIN ID: ZDB-GENO -990520–2 | | |
| Genetic reagent (*Danio rerio*) | *huli hutu* | *Mohideen et al., 2003* | | |
| Chemical compound, drug | 10% neutral buffered formalin | Fisher Scientific | | |
| Chemical compound, drug | EMBed-812 | Electron Microscopy Sciences | | |
| Chemical compound, drug | ethyl alcohol | Pharmco-Aaper | | |
| Chemical compound, drug | Finquel (MS-222, tricaine-S) | Argent Chemical Laboratories | | |
| Chemical compound, drug | glycol methacrylate | Polysciences | | |
| Chemical compound, drug | Kapton tubing | Small Parts | | |
| Chemical compound, drug | phosphate-buffered saline | Sigma-Aldrich | | |
| Chemical compound, drug | phosphotungstic acid | VWR | | |
| Chemical compound, drug | Ovadine | Syndel | | |
| Software, algorithm | Avizo | Thermo Fisher Scientific | SCR_014431 | version 9.4 |
| Software, algorithm | Elastix | http://elastix.isi.uu.nl/ | SCR_009619 | version 4.8 |
| Software, algorithm | Fiji/ImageJ2 | https://fiji.sc/ | SCR_002285 | |
| Software, algorithm | Ilastik | *Sommer et al., 2011* (http://ilastik.org/) | SCR_015246 | version 1.3 |

*Continued on next page*

*Continued*

| Reagent type (species) or resource | Designation | Source or reference | Identifiers | Additional information |
|---|---|---|---|---|
| Software, algorithm | ITK-SNAP | *Yushkevich et al., 2006* ([http://www.itksnap.org](http://www.itksnap.org)) | SCR_002010 | version 3.4 |
| Software, algorithm | OpenSeaDragon | [https://openseadragon.github.io/](https://openseadragon.github.io/) | | |
| Software, algorithm | TomoPy | Argonne National Labs ([http://tomopy.readthedocs.io](http://tomopy.readthedocs.io)) | | |
| Software, algorithm | VGStudio Max 2.1 | Volume Graphics | | |

## Zebrafish husbandry and sample preparation

Wild-type zebrafish (Ekkwill strain) and the *huli hutu* mutant (*Mohideen et al., 2003*) were reared at an average temperature of 28°C in a recirculating system with a 14:10 hr light:dark cycle (*Copper et al., 2018*). Fish were fed three times a day a diet consisting of brine shrimp and flake food. All fish were staged according to the zebrafish developmental staging series of *Kimmel et al. (1995)*.

After staging, larval (2, 3, 4, and 5 dpf) and juvenile (33 dpf) zebrafish specimens were euthanized in pre-chilled 2x Finquel (MS-222 or tricaine-S, 400 mg/L) solution (Argent Chemical Laboratories, Redmond, WA) buffered in 1% phosphate-buffered saline (PBS), and fixed in chilled 10% neutral buffered formalin (NBF) (Fisher Scientific, Allentown, PA) overnight in flat-bottom containers at room temperature. To improve fixation and reduce the volume of gut contents, we starved juvenile fish for at least 24 hr for a 10 mm specimen. Fixed zebrafish specimens were rinsed 3 times in 1X PBS for 10 min followed by being submerged in 35% ethyl alcohol (EtOH) for 20 min at room temperature with gentle agitation. The samples were then submerged in 50% EtOH for 20 min at room temperature with gentle agitation. Specimens were stained with 0.3% phosphotungstic acid (PTA) (diluted from a 1% w/v stock solution at a ratio of 3:7 in 100% EtOH) for 24 hr at room temperature with gentle agitation. This metallic stain is used widely in histology and electron microscopy for staining collagen and other connective tissue (*Bloom and Aghajanian, 1968*; *Bulmer, 1962*). The complete infiltration and embedding protocol is described elsewhere (*Lin et al., 2018*). Briefly, after staining, each specimen was infiltrated by sequential submerging in increasing concentrations of EtOH (70%, 90%, 95%, and 100%) for 30 min intervals at room temperature with gentle agitation. The samples were then embedded in glycol methacrylate (Polysciences, Inc, Warrington, PA) or EMBed-812 (Electron Microscopy Sciences, Hatfield, PA) in Kapton tubing (Small Parts, Inc, Logansport, IN). The tubing provides structural support with high thermal stability. Total sample preparation time takes 5 days. All procedures on live animals were approved by the Institutional Animal Care and Use Committee (IACUC) at the Pennsylvania State University.

Generation of the ENU-mutagenized mutant *hht* was previously described (*Mohideen et al., 2003*). The *hht* line was maintained as heterozygotes due to the recessive larval-lethal nature of the mutation. Mating was carried out by placing male and female heterozygotes in Aquatic Habitat breeding tanks with dividers the afternoon prior to egg collection. Collected eggs were disinfected in 10% Ovadine (Syndel) for 1 min at room temperature followed by three washes with charcoal-filtered water. Larvae were incubated at 28.5°C to maintain consistent speed of development. Homozygous mutant larvae were identified by a combination of gross phenotypes, including small eyes, small head, dorsally curved body, and enlarged yolk, that are easily detected under a low-power stereomicroscope at ≥3 dpf. Gross mutant phenotype at two dpf is limited to small eyes and requires screening at 40X.

## Image acquisition

Synchrotron micro-CT studies were performed on the beamline 2-BM-B Advanced Photon Source at Argonne National Laboratory. The beamline's quasi-parallel X-rays were used to acquire projection images of larval and juvenile zebrafish in sets of 2048 digital slices. After passing through the object, the X-rays impinge on a thin scintillator, which converts X-rays to optical photons that are magnified

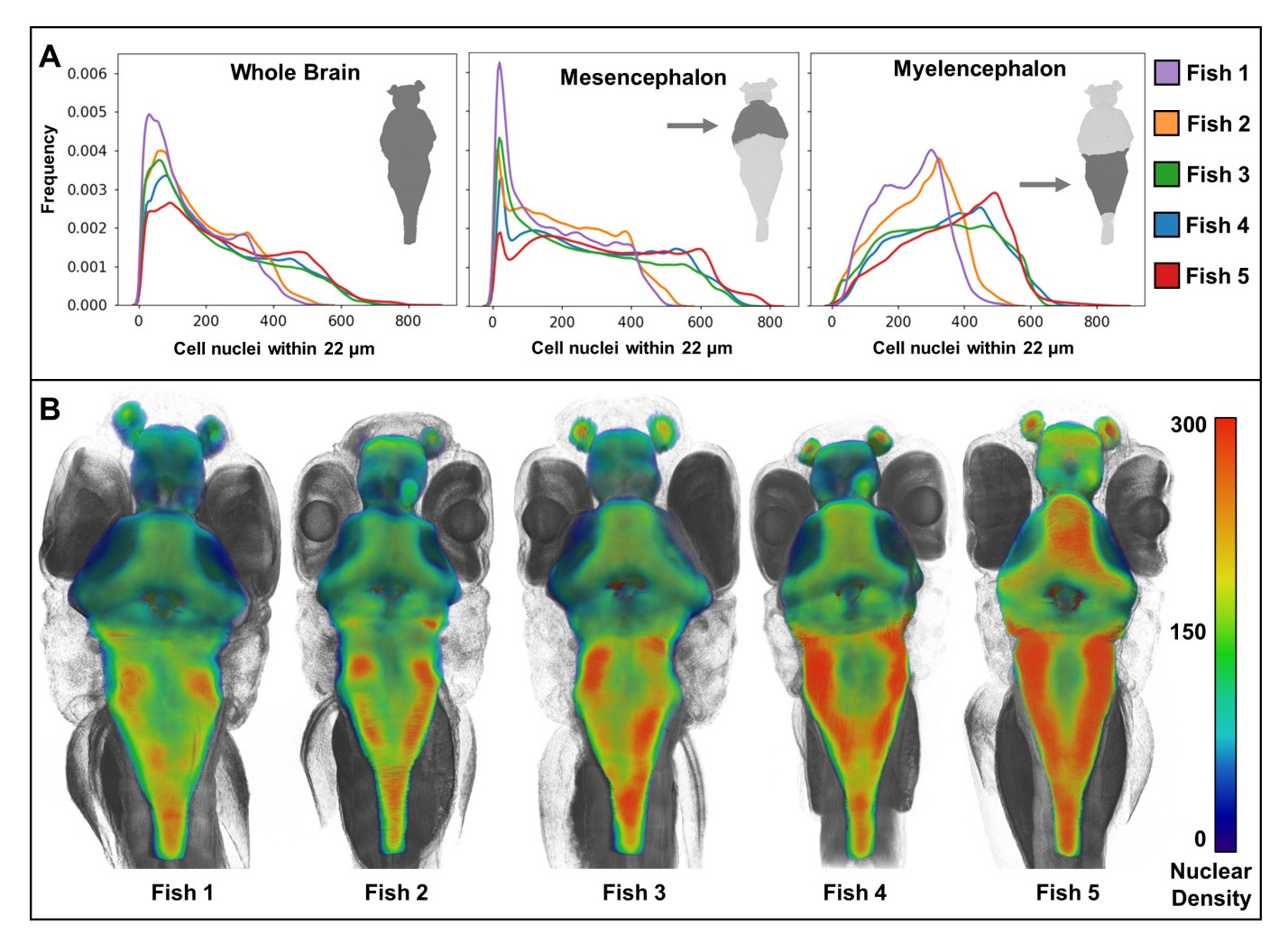

**Figure 9.** Heat maps of regional density of cell nuclei reveal striking phenotypic variation between larval zebrafish. Cellular density varies between individual brain regions and carries distinct signals consistent between individual samples (**A**). 3D renderings of whole-brain densities with identical transparency settings are presented for each fish (**B**) and reflects signal differences presented in (**A**).
DOI: https://doi.org/10.7554/eLife.44898.025

by a microscope objective lens onto a cooled CCD. The volumetric field of view was 1.5 mm³ for 0.743 μm³ voxels using a 10X objective lens, and 3 mm³ with 1.43 μm³ voxels using a 5X objective. A vertical stage was used to translate the sample between multiple acquisitions.

## Monochromatic imaging

For monochromatic imaging, a multilayer monochromator with bandwidth ΔE/E ~ 1.5%, or 200–250 eV, was used to obtain discrete X-ray energies. 1501 projections were obtained over 180 degrees (1 projection every 0.12 degrees) with a 2048-by-2048 pixel CoolSnap HQ CCD camera (Photometrics, AZ, USA). Larval and juvenile zebrafish scans were obtained at 13.8 keV and 16.2 keV, respectively. Additionally, two flat-field (gain) images (one at the beginning and one at the end of the acquisition) and one dark-field image are also acquired. Flat-field and dark-field corrections, ring artifact reduction, and image reconstruction were done using the open source TomoPy toolkit. Reconstructing the projection data generates a 3D data set comprised of a 2048-by-2048-by-2048 voxel cube. A voxel in the PTA-stained larval zebrafish has nominal 0.743 μm x 0.743 μm x 0.743 μm resolution, while a voxel in the juvenile zebrafish has a nominal 1.43 μm x 1.43 μm x 1.43 μm resolution, corresponding with larval and juvenile zebrafish fields-of-view of 1.5 mm x 1.5 mm x 1.5 mm and 3 mm x

3 mm x 3 mm, respectively. Whole-organism zebrafish reconstructions were created by combining series of segmental reconstructions at each vertical position along the full length of each specimen. Exposures of 400 ms per projection yielded acquisitions of ~20 min. Whole zebrafish scans take 3–5 acquisitions, depending on asample size.

## Polychromatic imaging

A polychromatic 'pink-beam' covering 10–30 keV, was used to obtain 1600 projections over 192 degrees (1 per 0.12 degrees) with a 2048-by-2048 pixel CoolSnap HQ CCD camera. Whole-organism imaging polychromatic reconstructions were generated from segmental reconstructions over its full length, taking ~20 s per segment. Stripes present in the raw pink-beam projection data were removed using a Fourier-Wavelet based method with a Haar filter and sigma of 3 pixels (*Münch et al., 2009*). Rings and bands were corrected in the polar transform domain of the reconstructed images. Free parameters in this method were optimized by using minimum entropy approaches.

## Estimates of the suitability of synchrotron micro-CT for phenome projects

The acquisition times using monochromatic or polychromatic 'pink-beam' sources are about 20 min or 20 s per scanned segment, respectively. Assuming: (i) a 5 min set up time per specimen, (ii) 10 replicates per condition, (iii) three scans per fish, (iv) a synchrotron availability of 100 scanning days per year and 10 hr of scanning time per scanning day, and (v) ability to multiplex five larval specimens at a time without increasing acquisition time, screening 20,000 larval mutants would require ~40 years with monochromatic source or ~1 year using pink-beam. There is potential for higher throughput via robotic sample swapping for monochromatic and polychromatic sources and/or increased multiplexing with pink-beam. Estimates of throughput would also be affected by the proportion of specimens that are juvenile or adult, since those are likely to be imaged individually, and require either multiple segmental scans per fish or spiral CT as used for humans.

## Energy optimization

In X-ray based imaging modalities, including micro-CT, the choice of imaging energy determines both the contrast between different tissue types and the noise level in the image. The contrast is based on the difference in energy-dependent absorption coefficient between the materials of interest, and the noise is determined by the transmitted flux of X-rays. In medical imaging, the contrast is often maximized at lower energies, for which differential absorption is maximized, while relative noise is reduced at higher energies, for which transmitted flux is maximized. A contrast-to-noise ratio

**Table 3.** Brain Volumes by Region for five dpf Zebrafish Brain.
Major divisions of the zebrafish brain and their volumes ($\mu m^3$) are shown across five zebrafish. Mean total volume is 15,507,461 and relative standard deviation ± 11.9%.

| Major brain divisions | Brain region volumes ($\mu m^3$) | | | | |
|---|---|---|---|---|---|
| Specimen number | 1 | 2 | 3 | 4 | 5 |
| Olfactory Epithelium | 293,251 | 171,459 | 164,338 | 131,459 | 111,195 |
| Telencephalon | 1,117,500 | 1,021,518 | 956,493 | 826,273 | 672,572 |
| Diencephalon | 1,440,447 | 1,300,518 | 1,097,698 | 1,134,753 | 1,046,486 |
| Hypothalamus | 813,210 | 702,116 | 732,073 | 608,018 | 575,387 |
| Mesencephalon | 4,353,108 | 3,442,415 | 3,663,909 | 3,183,413 | 2,885,684 |
| Metencephalon | 324,562 | 273,699 | 388,409 | 529,656 | 296,336 |
| Myelencephalon | 4,982,069 | 4,636,451 | 3,977,877 | 4,550,207 | 3,850,373 |
| White Matter | 4,783,477 | 4,358,188 | 4,119,302 | 3,963,126 | 3,529,787 |
| Spinal Cord | 78,092 | 97,609 | 117,260 | 109,808 | 125,722 |
| Total | 18,185,716 | 16,003,973 | 15,217,359 | 15,036,713 | 13,093,542 |

DOI: https://doi.org/10.7554/eLife.44898.027

(CNR) is generally maximized at some intermediate energy that depends on the concentration and amounts of various tissues present. Metal-stained tissue tomography is associated with the potential addition of X-ray absorption edges introducing additional structure into the energy-dependence of attenuation coefficients.

To investigate these tradeoffs, we made use of a simple but powerful model from *Spanne (1989)* that assumes the presence of a small circular contrasting detail at the center of a circular absorbing background. The background linear attenuation coefficient for energy $E$, $\mu_1(E) = \mu_{bg}(E)$. The attenuation coefficient for the contrasting material $\mu_2(E) = \mu_{bg}(E) + \mu_c(E)$. Note that each attenuation coefficient is given by the product of the material density in g/cm$^3$ and of a tabulated mass-attenuation coefficient in cm$^2$/g, that is $\mu_c(E) = \rho_c \left[\frac{\mu}{\rho}\right]_c (E)$.

The contrast detail is assumed to be large enough that blurring during reconstruction does not bias the reconstructed attenuation coefficient from its true value. The reconstructed noise variances at the center of the image in the presence and absence of the contrast detail are denoted $var\{\mu_1(E)\}$ and $var\{\mu_2(E)\}$, respectively. Because of circular symmetry, the variance at the center of a reconstructed image can be calculated in closed form (*Kak and Slaney, 1988*) and shown to be inversely proportional to the transmitted intensity.

$$var\{\mu_i(E)\} \propto \frac{1}{\bar{I}_i(E)} \,,$$

where $\bar{I}_1(E) = I_0(E)\exp\{-\mu_{bg}(E)t_{bg}\}$ and $\bar{I}_2(E) = I_0(E)\exp\{-\left[\mu_{bg}(E)d_{bg} + \mu_c(E)d_c\right]\}$ are the transmitted intensities in the absence and presence of the contrast detail, respectively. Here $I_0(E)$ is the incident intensity at energy $E$ and $d_c$ and $d_{bg}$ are the diameters of the contrast detail and background circle, respectively. With this model, we can then readily calculate a CNR defined as

$$CNR(E) = \frac{|\mu_1(E) - \mu_2(E)|}{\sqrt{var\{\mu_1(E)\} + var\{\mu_2(E)\}}}.$$

We used this model to calculate CNR ratios for the PTA stain as a function of background material thickness and tungsten contrast detail concentration. These are shown in *Figure 1—figure supplement 1* and both sets of plots show a clear optimal energy range just above the tungsten L1 edge at 12 keV with strong fall off below the L3 edge at 10.2 and again above about 16 keV. We thus chose to acquire all PTA stained larval data at the pre-calibrated 13.8 keV monochromator setting. For larger juvenile fish of diameter >3 mm, the CNR peak is broader, justifying the use of higher, more penetrating X-ray energies. Therefore, we chose the 16.2 keV monochromator setting for juvenile specimens.

## Phase contrast optimization

The coherence of synchrotron radiation allows for imaging that is sensitive to phase shifts in the incident X-ray wave front caused by variations in the real part of the complex index of refraction. A variety of approaches have been explored for performing quantitative phase-contrast tomography involving the use of grating analyzers or multiple measurements with different sample-scintillator distances (*Paganin, 2006*). The latter techniques rely on the fact that interference fringes develop in the transmitted intensity pattern as it propagates beyond the sample. Here, we were seeking a degree of edge perception resembling that seen in histology rather than quantitative phase-contrast. Juvenile (33 dpf) reconstructions at sample-to-scintillator distances (SSDs) of 10, 40, and 80 mm are reported in *La Rivière et al. (2010)*. Herein, we generated larval (five dpf) reconstructions at SSDs of 20, 30, 40 and 50 mm. We focused on the zebrafish eye to take advantage of the periodicity of retinal photoreceptors (*Figure 1—figure supplement 2*). The SSD of 20 mm caused edges of nuclei to appear blurry. SSDs of 30 or 40 mm yielded a degree of edge perception resembling that achieved in glass slide histology. Subjectively SSDs of 50 mm and larger (data not shown) caused edge effects to begin to look 'artificial' compared with traditional histology and transmission electron microscopy. Exaggerated phase effects can diminish perceived resolution. Notably, an 'ideal' SSD depends upon the specific structures being defined. Line profiles through the periodic retinal cells show that higher SSDs give rise to line profiles with a higher modulation depth, with the greatest benefit achieved around 30 and 40 mm SSD and strong evidence of overshooting and bias above

50 mm SSD. Based on these considerations, we used an SDD of 30 mm for all subsequent acquisitions.

## Image reconstruction, Image Processing and Visualization

CT reconstruction was done using TomoPy, an open-source package from Argonne National Labs (http://tomopy.readthedocs.io). Some of the image processing was conducted in Fiji/ImageJ2 (https://fiji.sc/). Software used for volumetric visualizations in this manuscript include Avizo version 9.4 (Thermo Fisher Scientific, Waltham, MA) and VGStudio Max 2.1 (Volume Graphics, Heidelberg, Germany). *Videos 1* and *4* were generated in Avizo. *Videos 2*, *3*, *5* and *6* were generated in VGStudio Max 2.1 using an intensity histogram adjusted to better discern otherwise faint or overlapping structures.

## ViewTool

The file sizes associated with synchrotron micro-CT are on the order of ~100 GB per larval or juvenile zebrafish, making scans difficult to view for people with standard computational resources. To allow users to inspect the data without having to download the full resolution volumes, we developed ViewTool, an open-access and web-based multiplanar viewer (http://3D.fish) (*Figure 5—figure supplement 1*). ViewTool is partly based on the open-access project OpenSeaDragon (https://openseadragon.github.io/), and combines radiology and digital pathology workflows into a seamless experience to provide user-friendly access to our 3D data. Orthogonal image z-stacks are downsampled using JPG compression before being served to end users through Amazon Web Services' Simple Storage Service (AWS 3). For bandwidth considerations, only every fourth slice is currently shown in the viewer by default. These images are presented in three windows in a user's browser that are spatially linked to the user's mouse or multi-touch interface. The two planes orthogonal to the interrogated plane sync to the corresponding x,y mouse location. ViewTool's code was written in client-side JavaScript, HTML and CSS, requires no download, and has no server requirement to run the basic implementation.

## Landmark-based Cross-Atlas registration

Semi-automated segmentation was performed using Elastix version 4.8, an open-source registration software (Klein 2010, http://elastix.isi.uu.nl/). All registrations were performed in two parts: an affine registration for optimizing initialization positions of both images followed by a landmark based thin-plate spline registration. Landmarks were manually marked on corresponding anatomical regions. The detected brain regions were derived from segmented volumes of the larval (4 dpf) reference fish from the ViBE-Z zebrafish brain atlas (*Ronneberger et al., 2012*), which was used for the primary registration onto one of our 5 dpf samples. Validation of region detection was performed by cross-referencing to a histology-based developmental brain atlas (*Wullimann and Mueller, 2005*) and manual segmentation was used to correct areas of inaccurate registration. This process generated a foundation for the detection of brain regions in our remaining larval (5 dpf) specimens (n = 5 total samples). All other registrations used this segmented 5 dpf sample as the moving image for better detection accuracy with less extensive manual segmentation. Manual segmentations were performed using the open-source software, ITK-SNAP version 3.4 (*Yushkevich et al., 2006*) (http://www.itksnap.org). The ViBE-Z database and atlas are publicly available (http://vibez.informatik.uni-freiburg.de/) (*Ronneberger et al., 2012*).

## Cell detection and counting

Using a supervised learning approach, we distinguished brain cell nuclei from background by training examples using Ilastik version 1.3 (*Sommer et al., 2011*), a simple, user-friendly, open-source tool for interactive image classification, segmentation and analysis (http://ilastik.org/). Specifically, three 75 $\mu m^3$ regions were selected for training across the larval (5 dpf) zebrafish brain and notochord regions. Nuclei were manually segmented in 2D in each of three orthogonal views (sagittal, coronal, and axial). Features that were used in the classification including intensity (pixel value with various smoothing), edges (gradient, Laplacian of Gaussian, and difference of Gaussians) and texture (structured tensor of eigenvalues, Hessian of Gaussian eigenvalues). The random forest classifier assigned a probability of being a nucleus to any given pixel. We combined probabilities assigned

from each orthogonal plane to obtain the probabilities in a given volume, $P_{total}(x, y, z) = P_{coronal}(x, y, z) \times P_{saggital}(x, y, z) \times P_{transverse}(x, y, z)$. Original data were thresholded to show approximate locations of nuclei and their probabilities. The results of nuclear labeling based on the nuclear training set demonstrate that, as expected, brain cell nuclei have high probabilities, while blood vessels and background have low probabilities. From these measures, a probability threshold and size filter (80% or higher probability containing at least 8 voxels,~3.5 μm$^3$) were used to detect individual cell nuclei. Cell nuclei in the same training regions were manually segmented in order to assess the accuracy of the automated segmentation results and to justify the probability threshold and size filter settings. Automatically detected cell nuclei were checked against those manually detected and categorized as true-positive (*manual and automatic detected*), false negative (*manually detected but automatically undetected*), or false positive (*manually undetected but automatically detected*). F1 score was optimized over different settings to balance precision and recall between manual and automated segmentation.

## Resource sharing

ViewTool is publically available (http://3D.fish). Digital histology is publicly available from our Zebrafish Lifespan Atlas (http://bio-atlas.psu.edu) (*Cheng, 2004*). Registered and unregistered 8-bit reconstructions of the heads of five zebrafish larvae involved in analysis are available on the Dryad Digital Repository (https://doi.org/10.5061/dryad.4nb12g2), along with scripts written for cell nuclei detection, analysis, and sample registration. Other code used for analysis can be also be downloaded from the Dryad repository. Full resolution scans, including raw projection data, are available from researchers upon request as a download or by transfer to physical media.

## Acknowledgements

The authors are grateful for the unusually extensive and sustained collaborative effort required across disciplines and institutions, including that of technical and administrative support staff. We thank Steven Peckins and Drs. Timothy Cooper and Gordon Kindlemann for critical insight in the early stages of the project's conception and development. This paper is dedicated to the memory of Dr. Betty PT Cheng. The investigators acknowledge NIH funding support (PI: KCC, R24-RR017441, and PI: KCC, R24-OD018559), pilot award funding to KCC from the Huck Institutes of the Life Sciences and the Institute for Cyber Science, PSU, and support from the Pennsylvania Tobacco Fund and Penn State Department of Pathology for the Penn State Functional Genomics Core. The use of the APS, an Office of Science User Facility operated for the US Department of Energy (DOE) Office of Science by Argonne National Laboratory, was supported by the US DOE under contract no. DE-AC02-06CH11357. The authors, and not the funding agencies, are solely responsible for the content of this work.

## Additional information

### Funding

| Funder | Grant reference number | Author |
|---|---|---|
| NIH Office of the Director | R24-OD018559 | Patrick La Riviere<br>Keith C Cheng |
| National Institutes of Health | R24-RR017441 | Patrick La Riviere<br>Keith C Cheng |
| Huck Institutes of the Life Sciences | Pilot award funding | Keith C Cheng |
| Institute for Cyber Science, PSU | Pilot award funding | Keith C Cheng |
| Jake Gittlen Memorial Golf Tournament | Pilot award funding | Keith C Cheng |
| Pennsylvania Tobacco Fund | Penn State Zebrafish Functional Genomics Core | Keith C Cheng |

| Huck Institutes of the Life Sciences | VIrtual Slide Scanner | Keith C Cheng |

The funders had no role in study design, data collection and interpretation, or the decision to submit the work for publication.

### Author contributions

Yifu Ding, Data curation, Software, Formal analysis, Validation, Investigation, Visualization, Methodology, Writing—original draft, Writing—review and editing; Daniel J Vanselow, Resources, Data curation, Software, Formal analysis, Supervision, Validation, Investigation, Visualization, Methodology, Writing—original draft, Writing—review and editing; Maksim A Yakovlev, Software, Formal analysis, Validation, Investigation, Visualization, Methodology, Writing—original draft, Writing—review and editing; Spencer R Katz, Formal analysis, Visualization, Writing—original draft, Writing—review and editing; Alex Y Lin, Resources, Data curation, Investigation, Methodology; Darin P Clark, Data curation, Investigation, Methodology; Phillip Vargas, Data curation, Software, Formal analysis, Validation, Investigation; Xuying Xin, Data curation, Visualization, Methodology; Jean E Copper, Resources, Data curation, Supervision, Visualization, Methodology, Project administration; Victor A Canfield, Supervision, Writing—original draft, Writing—review and editing; Khai C Ang, Resources, Supervision, Writing—review and editing; Yuxin Wang, Conceptualization, Formal analysis, Supervision, Funding acquisition, Validation, Investigation, Methodology, Writing—review and editing; Xianghui Xiao, Resources, Supervision, Methodology; Francesco De Carlo, Resources, Supervision, Funding acquisition, Methodology; Damian B van Rossum, Formal analysis, Supervision, Validation, Writing—original draft, Project administration, Writing—review and editing; Patrick La Riviere, Keith C Cheng, Conceptualization, Resources, Data curation, Software, Formal analysis, Supervision, Funding acquisition, Validation, Investigation, Visualization, Methodology, Writing—original draft, Project administration, Writing—review and editing

### Author ORCIDs

Yifu Ding (iD) https://orcid.org/0000-0002-4629-5858
Daniel J Vanselow (iD) https://orcid.org/0000-0002-9221-8634
Maksim A Yakovlev (iD) https://orcid.org/0000-0003-1846-3751
Spencer R Katz (iD) http://orcid.org/0000-0002-5586-3562
Alex Y Lin (iD) https://orcid.org/0000-0002-1653-4168
Victor A Canfield (iD) http://orcid.org/0000-0002-4359-1790
Khai C Ang (iD) https://orcid.org/0000-0001-7695-9953
Patrick La Riviere (iD) https://orcid.org/0000-0003-3415-9864
Keith C Cheng (iD) https://orcid.org/0000-0002-5350-5825

### Ethics

Animal experimentation: All procedures on live animals were approved by the Institutional Animal Care and Use Committee (IACUC) at the Pennsylvania State University, ID: PRAMS201445659, Groundwork for a Synchrotron MicroCT Imaging Resource for Biology (SMIRB).

### Decision letter and Author response

Decision letter https://doi.org/10.7554/eLife.44898.034
Author response https://doi.org/10.7554/eLife.44898.035

## Additional files

### Supplementary files

• Transparent reporting form
DOI: https://doi.org/10.7554/eLife.44898.028

## Data availability

ViewTool is publicly available (http://3D.fish). Digital histology is publicly available from our Zebrafish Lifespan Atlas (http://bio-atlas.psu.edu) (Cheng, 2004). Registered and unregistered 8-bit reconstructions of the heads of five zebrafish larvae involved in analysis are available on Dryad (https://datadryad.org/) along with scripts written for cell nuclei detection, analysis, and sample registration. Full bit-depth scans, including raw projection data, are available from researchers upon request as a download or transfer to physical media. Due to the large size of these files, use of a traditional repository at the time of publication was impractical.

The following dataset was generated:

| Author(s) | Year | Dataset title | Dataset URL | Database and Identifier |
|---|---|---|---|---|
| Ding Y, Vanselow D, Yakovlev M, Katz S, Lin A, Clark D, Vargas P, Xin X, Copper J, Canfield V, Ang K, Wang Y, Xiao X, Carlo FD, Rossum Dv, Riviere PL, Cheng K | 2019 | Data from: Computational 3D histological phenotyping of whole zebrafish by X-ray histotomography | https://dx.doi.org/10.5061/dryad.4nb12g2 | Dryad Digital Repository, 10.5061/dryad.4nb12g2 |

The following previously published dataset was used:

| Author(s) | Year | Dataset title | Dataset URL | Database and Identifier |
|---|---|---|---|---|
| Ronneberger O, Liu K, Rath M, Rueß D, Mueller T, Skibbe H, Drayer B, Schmidt T, Filippi A, Nitschke R, Brox T, Burkhardt H, Driever W | 2012 | ViBE-Z: A Framework for 3D Virtual Colocalization Analysis in Zebrafish Larval Brains | http://vibez.informatik.uni-freiburg.de/ViBE-Z_96hpf_v1.h5 | ViBE-Z, ViBE-Z_96hpf_v1.h5 |

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
