## [Decision Letter]

Thank you for submitting your article "Computational 3D histological phenotyping of whole zebrafish by X-ray histotomography" for consideration by *eLife*. Your article has been reviewed by two peer reviewers, and the evaluation has been overseen by a Reviewing Editor and Didier Stainier as the Senior Editor. The following individual involved in review of your submission has agreed to reveal their identity: Stephen C Ekker (Reviewer #2).

Overall, the reviewers were highly enthusiastic about the utility of this method and its description in this manuscript. There are some relatively minor comments/requests, as outlined below, that would help make this paper even more useful.

*Reviewer #1:*

Ding et al. set out to use X-ray micro tomography (micro-CT) for histological studies of both zebrafish larvae and juveniles. The resolution and clarity of them images provided are outstanding and truly state-of-the-art. The ability to dive in and examine any stage and tissue up to 33 day post fertilization is incredibly powerful at the level where one could practically count all the cells of a juvenile fish. This manuscript is a very well written, comprehensive, and persuasive demonstration of an approach to 3D imaging of juvenile zebrafish that is best suited to high-throughput anatomical phenotyping. The method described therein combines sub-micron resolution that allows determination of the cellular structure within tissues with a 3D map of the whole organism for context. A search for Micro-CT in zebrafish on PubMed yields 22 results, but only 3 publications discuss use of this technique on soft tissue. This manuscript represents significant advances over all three in terms of resolution, image quality, and demonstrated ability to gather quantitative data from the 3D images. There is compelling discussion of the idea that this technique holds significant advantages over conventional histology, with the characterization of the pneumatic duct (subsection “Detection of Histopathological Features in Whole-Fish Histotomography”) being an especially convincing example. There is also ample, detailed justification for the parameters used in the CT imaging, which I believe will be especially helpful in a journal with a wide target audience who are largely unlikely to be experts in these techniques.

A minor concern is that two more citations may be appropriate, as only one of the previous papers dealing with soft-tissue imaging by micro-CT in zebrafish is discussed (Babaei, 2016). So that the reader may make comparisons between the data in this manuscript and previous work with zebrafish, it would be helpful to cite Delphine Cheng et al., 2016 for their 3D characterization of the zebrafish GI tract (at lower resolution) in the third paragraph of the Introduction. The 2015 paper by Seo and colleagues in Zebrafish presented work similar in scope to this manuscript and should be discussed. This group imaged whole juvenile zebrafish by synchrotron micro-CT with a pixel size of 0.65 μm (I am unsure of voxel size but the image resolution appears to be lower), using various stains to enhance contrast of soft tissue and blood vessels. (Despite the topics being similar, I think that the image quality in the current manuscript is much better, and the discussion of technique development is much more thorough.)

This manuscript is nothing short of a tour-de-force that highlights the micro-CT method. While the authors have not really applied this powerful new methodology to a specific biological problem, they did choose to characterize homozygous huli hutu (hht) mutants. hht was originally described as a homozygous lethal allele that produces "striking architectural and cytologic changes in several organs" and was mapped to chromosome 12 (Mohideen et al., 2003). This is a poorly understood mutation that to my knowledge has not been molecularly explained. The micro-Ct approached reveled an absence of the pneumatic duct although hht mutants clearly have unabsorbed yolk, small eyes and small heads that lack much of their telencephalon. This is a constellation of phenotypes widely observed in ENU screens of zebrafish that typically do not exhibit swim bladders (a structure inflated by these pneumatic ducts). So… I would say that this portion of the effort proves that you can apply the approach to characterize a mutation probably better than any prior method. Moreover, it demonstrates that if one had a mutant to study, and one wanted to really comprehensively characterize its phenotype, I can imagine no better histological method. However, this example hasn't forwarded our understanding of a specific biological problem. In short, this is really an editorial issue as to what *eLife* wants to require from a paper describing a methodical advance.

*Reviewer #2:*

In the paper, the authors present a new method for whole organism imaging at a micron-level precision using a novel microCT-based approach. This manuscript describes the technical details of this novel approach that required new imaging methods using a Synchrotron X-ray flux imaging system. This system is about 2000x faster than current commercial sources enabling the authors to conduct a series of new experiments. They use this to image zebrafish, Danio rerio, embryos and larvae gathering near-histological quality resolution in the entire animal. Effective informatics was also deployed such as semi-automated segmentation of the data. This method begins to address a major bottleneck in the zebrafish field – no comprehensive atlas. Already yielding interesting findings such as comprehensive phenotyping of a mutant or differential range of 'normal' wild type organisms. The idea that a substantial range of differential phenotypes are still called 'normal' is obvious to the first minutes of a human anatomy student's exposure in class; quantifying what's 'normal' for a model vertebrate such as zebrafish is an important baseline for many subsequent experiments of mutants. ViewTool is also an excellent resource to make existing imaging available for the zebrafish.

From a larger perspective, this method begins to address the issue that we don't know what we don't know. For example, when conducting analyses of mutants, most papers focus on one organ or structure based on a critical hypothesis that is being tested. But many mutants impact other organs or systems, and due to practical concerns, these are unaddressed. For work in specific fields, this is fine. However, this misses two major questions – first, how genes can have uniform impact despite playing a role in diverse systems, and second, when it comes to human health, we are talking about the whole person and often not just a singular issue in the clinic. Whole organism data begins to address both scientific questions.

The paper is well written and clearly articulated. The senior author has presented this work at many international meetings and has clearly incorporated peer review comments in preparing this submission.

Remaining questions that are not addressed in this current version of the manuscript involves long-term accessibility of the data and tool. For example, how can the dataset described here be used as a scaffold for a detailed morphological atlas for zebrafish? Despite the tools presented, what does it take for a lab conducting lightsheet imaging (as a use case example) to obtain the subcellular resolution imaged for potential overlay work? For deployment of this work on other mutants (or for other model systems), what will be needed?

---

## [Author Response]

Reviewer #1:[…] A minor concern is that two more citations may be appropriate, as only one of the previous papers dealing with soft-tissue imaging by micro-CT in zebrafish is discussed (Babaei, 2016). So that the reader may make comparisons between the data in this manuscript and previous work with zebrafish, it would be helpful to cite Delphine Cheng et al., 2016 for their 3D characterization of the zebrafish GI tract (at lower resolution) in the third paragraph of the Introduction. The 2015 paper by Seo and colleagues in Zebrafish presented work similar in scope to this manuscript and should be discussed. This group imaged whole juvenile zebrafish by synchrotron micro-CT with a pixel size of 0.65 μm (I am unsure of voxel size but the image resolution appears to be lower), using various stains to enhance contrast of soft tissue and blood vessels. (Despite the topics being similar, I think that the image quality in the current manuscript is much better, and the discussion of technique development is much more thorough.)

We agree with including discussion of the suggested articles in context. Resolution is commonly computed as the field-of-view divided by the number of pixels covering that field-of-view, assuming otherwise perfect optics from scintillator to imaging array. Experimental validation of image resolution requires the use of phantoms or other internal controls (in our case, striations of skeletal muscle with previously characterized resolution). It is worth noting that achieved resolution is frequently lower than computed reconstruction resolution.

Reviewer #2:

[…] Remaining questions that are not addressed in this current version of the manuscript involves long-term accessibility of the data and tool. For example, how can the dataset described here be used as a scaffold for a detailed morphological atlas for zebrafish? Despite the tools presented, what does it take for a lab conducting lightsheet imaging (as a use case example) to obtain the subcellular resolution imaged for potential overlay work? For deployment of this work on other mutants (or for other model systems), what will be needed?

Indeed, long-term accessibility of the data and tool is a critical factor needed to promote a broad variety of applications. We have now addressed this issue, in the Discussion of the manuscript.

The pancellular nature of X-ray histotomography makes it ideal for building atlases of normal at the cellular through organ level. A detailed morphological atlas will require full volume representations of fish including larval, juvenile and adult stages. These images will need to be accessible through web-based resources. Fulfilling the long-term goal of unbiased, computational phenotyping will depend on statistical definitions of normal, for both gross and microscopic anatomy. Recognition of “abnormal” phenotypes will be based on comparisons with the statistical normal.

For a lab conducting fluorescence-based imaging – light sheet imaging, for example, there are ways to obtain organismal context at cellular and subcellular resolution. Fluorescent images can be superimposed on a micro-CT atlas by registering one to the other in a region-specific manner. Alternatively, cell-to-cell correlations between fluorescent cells and their histotomographic counterparts in the same fish can be achieved by X-ray histotomography after fixation and metal staining of the fluorescently imaged samples.

Deploying this technology for model organism phenotyping across laboratories will benefit from standardization of sample preparation, imaging parameters, and methods of analysis. Synchrotron-based resources presently appear to be the most suitable for standardization due to their far greater through-put and superior image quality. The applicability of this technology across all tissue types indicates that dedicated synchrotron beamlines for histotomography are justified. Laboratory-based tissue micro-CT using commercial sources will vary in terms of resolution and image quality. For optimal cross-referencing across laboratories, these same samples can be re-imaged at a synchrotron-based resource and shared through a common repository.

To facilitate access to our data, we have uploaded relevant data sets (along with code, supporting data, and descriptions) onto the Dryad Digital Repository, available at https://doi.org/10.5061/dryad.4nb12g2, as suggested by *eLife*. In the future, we envision a community driven and supported common repository for these types of large imaging data sets, where visualization tools, like ViewTool, are built-in and used for data exploration and evaluation.